# Scaling, Benchmarking, and Reasoning of Vision-Language Agents for Mobile GUI Navigation

**Heng Qu** [* 1]  **Yike Liu** [* 2]  **Renren Jin** [2]  **Wenzong Zhang** [2]  **Pengzhi Gao** [2]  **Wei Liu** [2]  **Jian Luan** [2]

## Abstract

Vision–Language Models (VLMs) have shown rapid progress in mobile GUI navigation. This paper presents a systematic study of data scaling, benchmarking, and reasoning for VLM-based agents in this domain. To facilitate rigorous evaluation, we introduce `HyperTrack`[1], a large-scale dataset with over 16000 real-world tasks across more than 650 Chinese mobile applications, along with `GUIEvalKit`[2], an open-source toolkit for unified benchmarking of VLMs on offline GUI navigation tasks. Using HyperTrack, we analyze the effects of training data scale on both supervised and reinforcement-based finetuning. Our results show that reinforcement-based finetuning consistently outperforms supervised finetuning, particularly in out-of-domain settings, highlighting the synergy between data scaling and reinforcement learning. Leveraging GUIEvalKit, we further benchmark state-of-the-art (SOTA) VLMs and analyze how interaction history and reasoning capabilities influence task completion. Together, HyperTrack and GUIEvalKit provide a comprehensive platform for developing and evaluating VLM agents in mobile GUI navigation tasks.

## 1. Introduction

Graphical User Interfaces (GUIs) constitute the primary interface through which users interact with modern digital systems, particularly on mobile platforms. Enabling AI agents to navigate complex GUIs remains a fundamental challenge, as it requires integrating multimodal perception, long-horizon reasoning, and sequential decision-making. Recent advances in Vision–Language Models (VLMs) have sub-

stantially expanded the capabilities of such agents, demonstrating strong effectiveness in practical scenarios such as web browsing (Lù et al., 2024; Gou et al., 2025), software control (Liu et al., 2024; Qin et al., 2025b), and gaming (Wang et al., 2025), opening new opportunities for intelligent human–computer interaction.

Despite this progress, several challenges limit the development and evaluation of VLM-based agents for mobile GUI navigation. Existing datasets are relatively small in scale and predominantly focus on English-language applications, offering limited coverage of Chinese mobile apps and restricting rigorous, reproducible evaluation. Moreover, the effects of data scaling on agent performance remain poorly understood under both supervised and reinforcement-based fine-tuning paradigms. In addition, there is a lack of standardized evaluation tools that enable consistent benchmarking across diverse GUI tasks. Finally, how reasoning capabilities influence task success and generalization in mobile GUI navigation has not been systematically studied.

To address these gaps, we present a comprehensive study of data scaling, benchmarking, and reasoning in VLM-based agents for mobile GUI navigation. We introduce `HyperTrack`, a large-scale dataset comprising over 16000 real-world tasks across more than 650 Chinese mobile applications, covering diverse user interactions and interface designs. We also develop `GUIEvalKit`, an open-source toolkit that enables unified offline benchmarking of GUI agents. Leveraging `HyperTrack`, we systematically investigate how scaling the number of training demonstrations affects agent performance under both supervised and reinforcement-based finetuning, across in-domain and out-of-domain evaluation settings. In addition, using `GUIEvalKit`, we benchmark SOTA open-source VLMs and analyze how their reasoning capabilities influence task completion, providing insights into the factors that drive performance in mobile GUI navigation. The contributions of this paper can be summarized as follows:

- We provide a large-scale, diverse dataset and a unified evaluation toolkit, enabling rigorous benchmarking of VLM agents in mobile GUI navigation.

- We systematically study how supervised and DAPO-

---

[1]Wuhan University [2]Xiaomi. Correspondence to: Pengzhi Gao <gaopengzhi@xiaomi.com>.

*Proceedings of the 43ʳᵈ International Conference on Machine Learning*, Seoul, South Korea. PMLR 306, 2026. Copyright 2026 by the author(s).

[1]HyperTrack preview subset; see Appendix A.2.
[2]https://github.com/xiaomi-research/guievalkit

| Dataset | Tasks | Apps or websites | Steps | Hier. docs. | Prec. bbox | Screen desc. | Instr. level | Lang. |
|---|---|---|---|---|---|---|---|---|
| AITW (Rawles et al., 2023) | 715142 | 357+ | 6.5 | ✗ | ✗ | ✗ | high | en |
| AndroidControl (Li et al., 2024) | 15283 | 833 | 5.5 | ✓ | ✗ | ✗ | high&low | en |
| AiTZ (Zhang et al., 2024) | 2504 | 70+ | 7.5 | ✗ | ✗ | ✓ | high&low | en |
| AMEX (Chai et al., 2025) | 3046 | 192 | 12.8 | ✗ | ✗ | ✓ | high | en |
| GUI Odyssey (Lu et al., 2025a) | 8334 | 212 | 15.3 | ✗ | ✗ | ✓ | high&low | en |
| CAGUI (Zhang et al., 2025b) | 600 | 22 | 7.5 | ✗ | ✗ | ✗ | high | zh |
| HyperTrack | 16080 | 674 | 5.1 | ✓ | ✓ | ✓ | high&low | zh |

*Table 1.* Comparison between HyperTrack and existing mobile GUI datasets in terms of: (i) number of task demonstrations, (ii) number of apps or websites, (iii) average task length, (iv) availability of hierarchical UI documents, (v) presence of action bounding boxes, (vi) screen descriptions, and (vii) task annotations using high-level or low-level instructions.

style reinforcement-based finetuning scale with training data, revealing the synergy between data volume and reinforcement learning.

- We benchmark state-of-the-art VLMs and analyze how interaction history and reasoning capabilities influence task completion, offering insights into the key factors driving GUI agent performance.

## 2. The `HyperTrack` Dataset

### 2.1. Dataset Description

The HyperTrack dataset is collected from 674 Chinese Android apps spanning 17 categories (Figure 9). It covers a diverse range of applications, including popular apps (e.g., Weibo, Taobao, Amap), long-tail less-popular apps, and tablet-specific apps. For each demonstration, annotators first provide a high-level task description in natural language, designed to capture the core functionality of the target app. During data collection, annotators interact with the device using a predefined set of actions: OPEN, CLICK, SCROLL, TYPE, and STOP. Each step is accompanied by a screen description and a low-level action description. For all clickable actions, annotators additionally annotate ground-truth bounding boxes corresponding to the executed actions. A comparison of HyperTrack with existing mobile GUI datasets is shown in Table 1, and detailed data collection process and dataset statistics are provided in Appendix A.

### 2.2. Data Scaling with the `HyperTrack` Dataset

We split HyperTrack into one training set, one validation set, and four test sets (Table 8). The test sets are designed to evaluate both in-domain and out-of-domain generalization: (1) in-domain, consisting of tasks sampled from the same distribution as the training data; (2) unseen app, containing tasks from apps not present in training; (3) unseen device, containing tasks collected on unseen devices; (4) unseen app & device, containing tasks from both unseen apps and devices. Note that the training set contains only mo-

bile phone demonstrations, while the unseen-device splits include tablet demonstrations.

To study the effect of data scaling, we create ten training subsets with 16, 32, 64, 128, 256, 512, 1024, 2048, 4096, and 8192 episodes. For each subset and the full training set, we finetune UI-TARS-1.5-7B (Qin et al., 2025a) using both supervised finetuning (SFT) and reinforcement-based finetuning (RL). For RL training, we adopt Group Relative Policy Optimization (GRPO) (Shao et al., 2024) with three enhancements from DAPO (Yu et al., 2025). Clip-Higher increases the upper clipping bound in the GRPO objective, Dynamic Sampling replaces zero-advantage samples with non-zero-advantage ones to improve the gradient signal, and Token-Level Policy Gradient Loss ensures that each token contributes equally to the gradient computation. Formally, let $\pi_\theta$ denote a policy parameterized by $\theta$. Given a prompt $q$, the policy generates $G$ responses $\{o^1, \ldots, o^G\}$ with corresponding rewards $\{R_1, \ldots, R_G\}$. Let $\epsilon_{\text{low}}$ and $\epsilon_{\text{high}}$ denote the lower and upper clipping bounds, respectively, $\beta$ the coefficient of the KL-divergence penalty, and $\mathcal{D}$ the training dataset of prompts. The GRPO objective is defined as:

$$\mathbb{E}_{q \sim \mathcal{D}, \{o^i\}_{i=1}^{G} \sim \pi_{\theta_{\text{old}}}(\cdot|q)} \left[ \frac{1}{\sum_{i=1}^{G} |o^i|} \sum_{i=1}^{G} \sum_{t=1}^{|o^i|} \min \Big( r_{i,t}(\theta) \hat{A}_{i,t}, \right.$$

$$\left. \text{clip}\left( r_{i,t}(\theta), 1 - \epsilon_{\text{low}}, 1 + \epsilon_{\text{high}} \right) \hat{A}_{i,t} \Big) - \beta \mathbb{D}_{KL}\left( \pi_\theta || \pi_{\text{ref}} \right) \right],$$

where $r_{i,t}(\theta) = \frac{\pi_\theta\left(o_t^i | q, o_{<t}^i\right)}{\pi_{\theta_{\text{old}}}\left(o_t^i | q, o_{<t}^i\right)}$, and the normalized advantage is computed as $\hat{A}_{i,t} = \frac{R_i - \text{mean}(\{R_1, \ldots, R_G\})}{\text{std}(\{R_1, \ldots, R_G\})}$. We fix $G$ to be 16. The clipping bounds $\epsilon_{\text{low}}$ and $\epsilon_{\text{high}}$ are set to 0.2 and 0.3, respectively. Following Zhou et al. (2025), we set $\beta$ to zero, which removes the need for a reference model during training and reduces GPU memory consumption.

The reward $R$ for each response is decomposed into an action-type reward $R_{\text{action-type}}$ and a parameter reward $R_{\text{params}}$. The action-type reward is binary, with $R_{\text{action-type}} = 1$ if the predicted action type matches the ground truth and

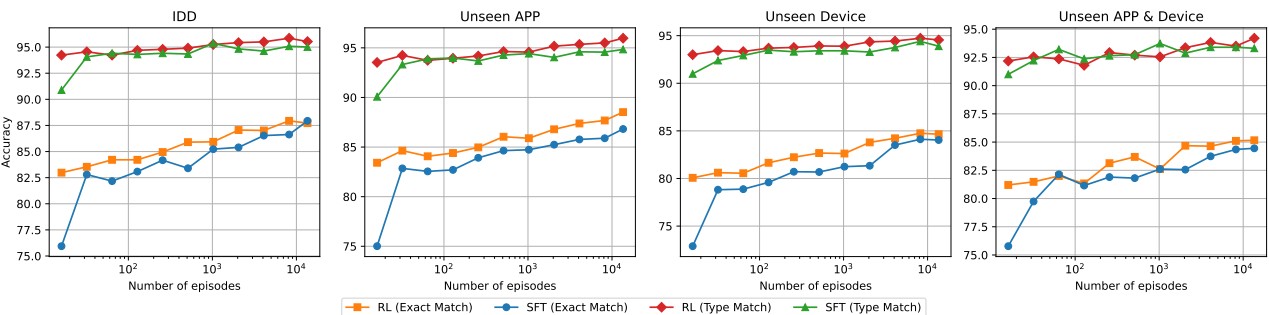

*Figure 1.* Comparison of Type Match and Exact Match across models trained with different numbers of episodes under supervised and reinforcement-based finetuning. Type Match is the proportion of responses whose predicted action type matches the ground truth, i.e., $R_{\text{action-type}} = 1$. Exact Match is the proportion of responses with a correct action type and parameters, i.e., $R_{\text{params}} = 1$.

0 otherwise. The parameter reward is evaluated only when the action type is correct; otherwise, $R_{\text{params}} = 0$. When applicable, $R_{\text{params}}$ is defined by the action type: for click actions, it is 1 if the predicted coordinate lies within the ground-truth bounding box; for scroll actions, it is 1 if the predicted direction matches the ground truth; and for text-argument actions (e.g., OPEN and TYPING), it is 1 if the generated text exactly matches the ground truth. The final reward is given by

$$R = R_{\text{action-type}} + R_{\text{params}}. \tag{1}$$

Figure 1 illustrates how model performance under SFT and RL varies with training data scale. We summarize the key observations below:

- Performance scales approximately linearly with the logarithm of training data size. Both SFT-trained and DAPO-style RL-trained models exhibit an approximately linear improvement in type and exact accuracy with respect to the logarithm of the number of training episodes.

- DAPO-style RL consistently outperforms SFT at the same data scale, with particularly strong gains in out-of-domain settings. While RL-trained models achieve only modest improvements over SFT-trained models in the in-domain setting, the performance gap becomes substantially larger in out-of-domain scenarios (e.g., the unseen app split), indicating superior generalization of RL-trained models.

To assess whether the observed scaling behavior depends on a specific backbone or reward formulation, we further conduct complementary experiments using Qwen3-VL-8B-Thinking under SFT and RL. In addition to the binary reward described above, we evaluate an alternative Gaussian spatial reward for click actions, motivated by Gaussian reward modeling for GUI grounding (Tang et al., 2025a). The Gaussian variant is applied only to click-coordinate prediction, where dense spatial feedback is most relevant, while all other action parameters continue to use the binary reward definition

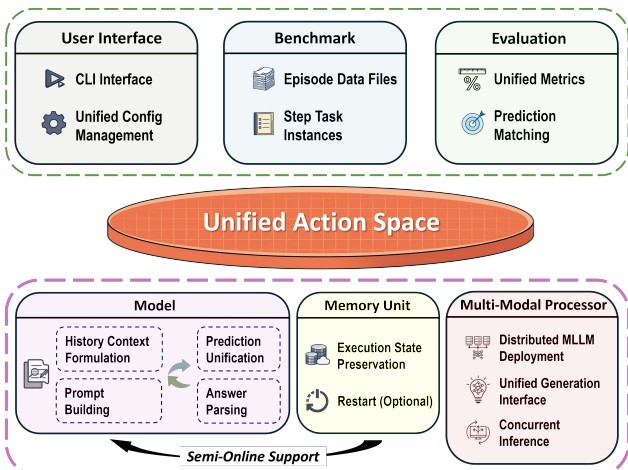

*Figure 2.* Main components in GUIEvalKit.

described above. Click actions account for 69.3% of the HyperTrack training set, making this setting representative of the dominant action category. The results show that the overall scaling trend remains consistent across backbones and reward formulations. RL continues to outperform SFT at comparable data scales, and the Gaussian reward achieves performance comparable to the binary reward while providing denser supervision for click actions. Appendix A.3 provides the detailed Qwen3-VL-8B-Thinking results under SFT, binary RL, and Gaussian spatial reward settings.

## 3. The `GUIEvalKit` Toolkit

Figure 2 illustrates the main components of GUIEvalKit, which are discussed in detail in the following sections.

### 3.1. Supported Benchmarks & Models

GUIEvalKit integrates five representative GUI benchmarks that together span diverse tasks, environments, and linguistic settings. AndroidControl and AiTZ focus on large-scale, semantically rich tasks in English-centric scenarios. GUI

Odyssey emphasizes structurally complex and compositional cross-app control, while CAGUI and HyperTrack extend evaluation to realistic Chinese-language GUI interactions. All benchmarks are normalized into a unified action space summarized in Table 9. Each episode step is encapsulated by a `StepTaskModel`, which stores immutable metadata (e.g., observations, instructions, and ground-truth actions) and runtime artifacts (e.g., predictions and evaluation results), providing a unified abstraction for task specification, evaluation, and result persistence.

GUIEvalKit currently supports eight model families, encompassing over 30 VLMs. These include general-purpose VLMs with GUI interaction capabilities, such as Qwen2.5-VL (Bai et al., 2025b), Qwen3-VL (Bai et al., 2025a), MiMo-VL (Team et al., 2025a), and GLM-4.1/4.5V (Team et al., 2025b), as well as GUI-specialized agents, including UI-TARS (Qin et al., 2025b), UI-Venus (Gu et al., 2025), GUI-Owl (Ye et al., 2025), MagicGUI (Tang et al., 2025b), and AgentCPM-GUI (Zhang et al., 2025a).

### 3.2. Model Inference

GUIEvalKit standardizes model inference through the `ABCModel` abstraction, which defines a uniform integration interface for all evaluated models. Integration begins with `.prepare_input()`, which formats task inputs into service-compatible messages. The inputs are then consumed by the unified `.generate()` method, which transparently adapts to different deployment backends. Raw model outputs are converted into structured actions using `.parse_response()`, producing the corresponding actions. An optional CLI flag, `enable_thinking`, enables automatic input adjustments for models supporting explicit reasoning traces, facilitating controlled evaluation of intermediate reasoning on GUI task performance.

Each `ABCModel` instance maintains a thread-safe memory unit that caches intermediate observations and textual context across steps, avoiding redundant data loading and enabling robust recovery from interrupted executions. GUIEvalKit supports both offline and online inference through mainstream frameworks such as vLLM (Kwon et al., 2023), with high-throughput evaluation enabled via Python concurrent execution. Parallelism, concurrency, and decoding parameters (e.g., temperature, top-$p$, top-$k$) are fully configurable to facilitate systematic experimentation.

### 3.3. Evaluation

#### 3.3.1. OFFLINE EVALUATION

The core evaluation objective across all GUI benchmarks is to measure how accurately a model reproduces valid execution trajectories. Each task is associated with at least one reference trajectory leading to a valid terminal state,

represented as a sequence of action–observation pairs:

$$\tau = ((a_0, o_0), (a_1, o_1), \ldots, (a_T, o_T)), \quad (2)$$

where $a_i$ denotes the reference action at step $i$ and $o_i$ the corresponding observation, typically a static screenshot. Offline evaluation proceeds step by step along this trajectory. At each step $i$, the model observes a reconstructed history:

$$h_i = \phi(\tau_{<i}) = (\phi(a_0, o_0), \ldots, \phi(a_{i-1}, o_{i-1})), \quad (3)$$

where $\phi(\cdot)$ adapts the interaction history to the model's input constraints, such as limits on visual context or intermediate reasoning traces. Each step is evaluated by comparing the model's predicted action to the reference action. Matching thresholds are managed by modular `ActionEvaluator` components, ensuring consistent evaluation across benchmarks. We report two step-level metrics (type match and exact match) and two episode-level metrics (progress and success rate) to capture long-horizon performance.

Note that benchmark and model action spaces may not fully overlap. To avoid unfairly penalizing models for predicting unsupported actions, we compute metrics only on steps where the predicted action exists in both the benchmark and model action spaces. For our evaluation results, we report metrics only for tasks where at least 95% of steps satisfy this condition. In addition, some ground-truth actions, such as OPEN, may be outside a model's native action space. To account for this, we also provide metrics computed after excluding these unsupported actions.

#### 3.3.2. SEMI-ONLINE EVALUATION

While efficient, offline evaluation relies on two assumptions. First, observations are static, and evaluation is conditioned on the model remaining on the reference trajectory. Second, the reconstructed history is off-policy and may omit model-specific decision artifacts, such as intermediate reasoning.

To reduce the mismatch between the evaluation context and the model's operational behavior while preserving the on-track assumption, we propose semi-online evaluation (SOEval). At each step $i$, if the model's predicted action $\hat{a}_i$ exactly matches the reference action $a_i$, we incorporate the model's own decision artifacts into the evaluation history; otherwise, we fall back to the reference trajectory. Formally, the history at step $i$ is defined as

$$(\psi(a_0, o_0, \hat{a}_0, \hat{\tau}_0), \ldots, \psi(a_{i-1}, o_{i-1}, \hat{a}_{i-1}, \hat{\tau}_{i-1})), \quad (4)$$

where $\hat{\tau}_t$ captures the model's reasoning process at step $t$, and the history selection operator $\psi(\cdot)$ is defined as

$$\psi(o_t, a_t, \hat{a}_t, \hat{\tau}_t) = \begin{cases} \varphi(o_t, \hat{a}_t, \hat{\tau}_t), & \text{if } \hat{a}_t = a_t, \\ \phi(o_t, a_t), & \text{otherwise.} \end{cases} \quad (5)$$

$\varphi(\cdot)$ selects a model-side representation that preserves on-policy decision artifacts. Under sustained on-track behavior, the semi-online history progressively approaches an ideal on-policy history composed of the model's own actions and reasoning artifacts. This allows SOEval to tighten the evaluation context by incorporating on-policy information when available, while preserving offline evaluation stability.

## 4. Evaluation Results

### 4.1. Baselines

We evaluate all models on the benchmarks supported by GUIEvalKit, using each model's recommended prompting format and decoding strategy during inference. For models that support controllable thinking modes, we report results both with and without the thinking mode enabled. Following prior work, we exclude the OPEN step from evaluation on the AndroidControl and HyperTrack benchmarks.

### 4.2. Main Results

**Offline Evaluation** Table 2 summarizes model performance across benchmarks under the offline evaluation setting. By comparing models of different sizes and inference configurations, we make the following observations:

- Specialized GUI agents (e.g., UI-TARS and AgentCPM-GUI) consistently outperform general-purpose VLMs on most benchmarks, highlighting the importance of task-specific training for robust GUI navigation.

- Larger model sizes consistently improve GUI navigation performance across both VLMs and GUI agents. For example, UI-TARS shows clear gains when scaling from 2B to 72B, indicating that larger models better capture long-horizon and compositional GUI interaction patterns.

- Thinking mode does not consistently improve performance and can be detrimental. For several models with controllable thinking modes, such as MiMo-VL and GUI-Owl, disabling thinking leads to higher step type and exact match. Similarly, for Qwen3-VL, the instruct variants outperform the thinking variants at both 4B and 8B scales. These results suggest explicit reasoning traces may introduce unnecessary verbosity or distract models from precise action execution in GUI navigation.

**Semi-Online Evaluation.** Table 3 shows the average step exact match performance across all supported benchmarks under the semi-online evaluation setting. Compared to offline evaluation, SOEval consistently improves PASS@1 performance across different model families and inference modes. These improvements primarily arise from evaluation context construction, as semi-online evaluation condi-

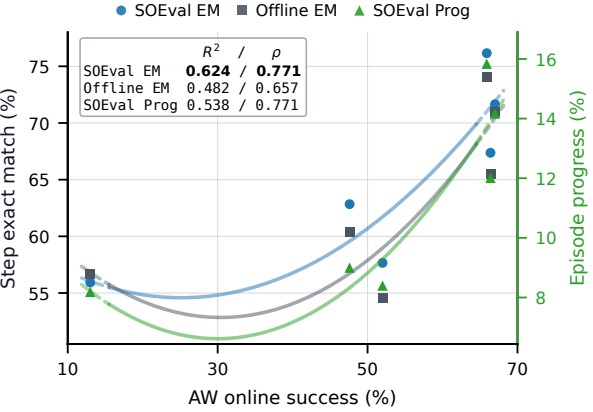

*Figure 3.* Correlation with AndroidWorld (AW) online success across six models. Left axis: SOEval/offline step exact match; right axis: SOEval progress; $\rho$: Spearman correlation; $R^2$: coefficient of determination. Curves show second-order Legendre fits. These empirical results support interpreting SOEval as a context-aligned, static-data, and step-level approximation, not as a substitute for full online evaluation.

tions models on self-generated interaction histories that are closely aligned with their own behavioral distributions.

To evaluate whether SOEval better reflects real interactive performance, we compare offline-style metrics with AndroidWorld (Rawles et al., 2025) online success across six representative models. As shown in Figure 3, SOEval step exact match exhibits a stronger correlation with AW online success than offline step exact match. SOEval episode progress also correlates with AW online success, although the results suggest that SOEval more reliably captures local decision quality than full episode execution dynamics. The raw correlation data are provided in Appendix D.1.

While the execution-level results in Tables 2 and 3 validate the above trends, aggregate evaluation metrics alone are insufficient to fully characterize model behavior. We therefore conduct a decision-level analysis in Section 5 to systematically investigate the following questions: 1) *How does SOEval improve evaluation context construction relative to offline evaluation?* 2) *How does explicit reasoning influence the model's decision-making behavior?*

## 5. Evaluation Analysis

### 5.1. Decision-Level Metrics

We introduce two decision-level analysis metrics, decision diversity and decision stability, to enable principled analysis of GUI agent behavior, revealing robustness, ambiguity, and exploration patterns that are not captured by standard step-level or episode-level evaluations.

For each step-level task $s$, we define a task-conditional

| Model | Android Control | | GUI-Odyssey | AiTZ | CAGUI | HyperTrack |
| | low-level | high-level | | | agent | |
| --- | --- | --- | --- | --- | --- | --- |
| **Open-Source VLMs** | | | | | | |
| Qwen2.5-VL-3B | 93.16 / 84.00 | 76.64 / 62.62 | 61.94 / 45.99 | 76.10 / 60.61 | 83.55 / 59.06 | 85.43 / 63.34 |
| Qwen2.5-VL-7B | 94.61 / 85.05 | 73.46 / 61.40 | 61.89 / 47.92 | 78.58 / 64.73 | 77.81 / 58.48 | 81.27 / 64.17 |
| GLM-4.1V-Thinking | 86.09 / 80.66 | 67.31 / 53.02 | 72.76 / 42.57 | 68.95 / 44.01 | 85.47 / 65.48 | 84.66 / 67.14 |
| GLM-4.5v | 86.35 / 81.37 | 71.54 / 59.15 | 75.33 / 48.90 | 70.72 / 48.52 | 87.64 / 69.26 | 86.76 / 72.97 |
| Qwen3-VL-4B-Thinking | 85.75 / 71.84 | 70.15 / 56.90 | 71.45 / 43.58 | 69.13 / 48.26 | 77.58 / 56.16 | 76.10 / 56.96 |
| Qwen3-VL-4B-Instruct | 82.15 / 72.59 | 71.39 / 58.68 | 78.01 / 50.36 | 75.20 / 52.57 | 82.80 / 62.32 | 81.81 / 65.81 |
| Qwen3-VL-8B-Thinking | 81.08 / 71.10 | 70.24 / 57.13 | 74.01 / 46.98 | 66.90 / 47.27 | 78.37 / 56.89 | 77.03 / 59.35 |
| Qwen3-VL-8B-Instruct | 82.36 / 72.20 | 71.82 / 59.95 | 77.85 / 51.78 | 72.77 / 52.64 | 83.83 / 63.94 | 81.48 / 66.28 |
| MiMo-VL-7B-SFT-2508 | 90.99 / 87.23 | 75.43 / 65.77 | 78.44 / 56.56 | 74.45 / 57.75 | 84.61 / 65.32 | 87.20 / 72.20 |
| w/o thinking | 92.84 / 89.07 | 79.02 / 68.98 | 89.45 / 70.31 | 81.69 / 66.83 | 83.08 / 63.80 | 93.13 / 77.98 |
| MiMo-VL-7B-RL-2508 | 91.93 / 87.78 | 77.92 / 68.66 | 79.82 / 59.72 | 75.19 / 60.22 | 86.49 / 67.96 | 88.96 / 74.89 |
| w/o thinking | 94.03 / 90.23 | 79.30 / 70.04 | 85.64 / 67.08 | 79.38 / 66.91 | 79.27 / 61.60 | 92.56 / 76.41 |
| **Open-Source GUI Agents** | | | | | | |
| UI-TARS-2B-SFT | 93.16 / 84.00 | 76.64 / 62.62 | 61.94 / 45.99 | 76.10 / 60.61 | 83.55 / 59.06 | 85.12 / 63.00 |
| UI-TARS-7B-SFT | 98.08 / 94.81 | 85.00 / 77.99 | 86.94 / 68.82 | 82.92 / 67.34 | 89.99 / 70.62 | **90.40** / 75.40 |
| UI-TARS-7B-DPO | 91.23 / 87.97 | 80.59 / 73.44 | 85.42 / 68.32 | 82.01 / 65.33 | 90.26 / 70.22 | 90.16 / 74.72 |
| UI-TARS-1.5-7B | 96.88 / 93.04 | 78.72 / 69.09 | 83.33 / 66.24 | 78.45 / 60.27 | 89.11 / 67.38 | 90.12 / 74.96 |
| UI-TARS-72B-SFT | **98.17 / 95.05** | **86.17 / 79.37** | 89.80 / 72.27 | 84.27 / 69.83 | 91.08 / 74.53 | 90.16 / 75.20 |
| UI-TARS-72B-DPO | 94.18 / 91.58 | 84.62 / 78.07 | 86.41 / 69.04 | 81.56 / 66.47 | 90.23 / 73.47 | 90.25 / **75.61** |
| AgentCPM-GUI-8B | 92.80 / 88.60 | 76.40 / 67.93 | **90.82 / 74.84** | **85.46 / 76.08** | **96.88 / 91.32** | 82.80 / 54.26 |
| MagicGUI-CPT | 88.91 / 81.19 | 78.50 / 67.39 | 88.84 / 74.70 | 65.45 / 44.90 | 72.05 / 46.15 | 85.56 / 65.06 |
| MagicGUI-RFT | 95.77 / 91.78 | 81.47 / 72.76 | 85.29 / 72.78 | 64.20 / 44.50 | 78.52 / 52.81 | 88.62 / 67.13 |
| UI-Venus-Navi-7B | 92.17 / 86.16 | 79.05 / 68.61 | 87.30 / 71.09 | 79.06 / 65.01 | 85.16 / 64.57 | 84.69 / 66.92 |
| UI-Venus-Navi-72B | 89.81 / 85.20 | 82.35 / 73.53 | 87.61 / 72.10 | 79.15 / 65.20 | 87.13 / 69.60 | 86.58 / 73.40 |
| GUI-Owl-7B | 83.06 / 77.01 | 81.52 / 72.01 | 78.67 / 61.26 | 74.53 / 60.33 | 82.44 / 60.25 | 80.67 / 63.99 |
| w/o thinking | 91.05 / 86.25 | 81.60 / 72.66 | 81.58 / 65.22 | 76.48 / 63.27 | 81.73 / 59.43 | 85.86 / 67.65 |
| GUI-Owl-32B | 80.86 / 76.08 | 83.59 / 75.59 | 84.05 / 70.10 | 73.09 / 58.64 | 83.99 / 64.99 | 82.79 / 67.09 |
| w/o thinking | 86.82 / 82.54 | 83.81 / 76.05 | 86.26 / 73.38 | 75.11 / 61.77 | 85.78 / 66.36 | 88.83 / 72.41 |

*Table 2.* The GUI navigation performance (step type match / step exact match) across different models under the offline evaluation settings. Note that "w/o thinking" denotes that we manually switch off the thinking mode.

| Model | Offline | SOEval |
| --- | --- | --- |
| Qwen2.5-VL-7B | 60.03 | 60.08 |
| Qwen3-VL-4B-Thinking | 54.52 | 54.84 |
| Qwen3-VL-4B-Instruct | 59.39 | 63.05 |
| Qwen3-VL-8B-Thinking | 54.98 | 57.39 |
| Qwen3-VL-8B-Instruct | 60.39 | 62.84 |
| GUI-Owl-7B | 65.49 | 67.37 |
| GUI-Owl-32B | 70.95 | 71.66 |
| UI-Venus-Navi-72B | 74.09 | 76.16 |

*Table 3.* Average step exact match across five benchmarks under offline and semi-online evaluation settings.

decision space $\mathcal{D} = \{d_1, \ldots, d_K\}$, where each element corresponds to a semantically distinct and valid decision at that step. This decision space serves as a shared semantic reference that is independent of the underlying model and sampling strategy. Let $\mathcal{E}$ denote the execution space of concrete action realizations. Given a model $\mathcal{M}$ under sampling configuration $\mathcal{S}$, we obtain $n$ execution samples $\{e_i\}_{i=1}^n$ via repeated rollouts. Since executions corresponding to the same decision are expected to be close under the action distance metric, we empirically approximate the execution-to-decision mapping by clustering sampled executions in $\mathcal{E}$ using density-based methods. Each cluster is treated as

a decision unit, denoted as $c(e_i)$, which induces a discrete decision distribution for model $\mathcal{M}$ on step task $s$:

$$p(d_k \mid \mathcal{M}, \mathcal{S}, s) = \frac{1}{n} \sum_{i=1}^n \mathbf{1}(c(e_i) = d_k), \qquad (6)$$

which captures how probability mass is allocated over semantic decisions. Comprehensive discussion of the clustering-based abstraction is provided in Appendix C.1.

**Decision Diversity.** Decision diversity measures how broadly a model explores alternative decisions under stochastic sampling. Formally, we define it as the entropy of the induced decision distribution $p(d \mid \mathcal{M}, \mathcal{S}, s)$:

$$
\begin{aligned}
\mathrm{Div}(\mathcal{M}, \mathcal{S}, s) &\triangleq H(p(d \mid \mathcal{M}, \mathcal{S}, s)) \\
&= -\sum_{k=1}^K p(d_k) \log p(d_k).
\end{aligned}
\qquad (7)
$$

Higher decision diversity indicates that the model assigns meaningful probability mass to multiple distinct decision options, reflecting either intrinsic ambiguity in the task or uncertainty in the model's decision-making process. In contrast, low diversity suggests more concentrated probability

mass and correspondingly more deterministic behavior.

**Decision Stability.** Decision stability measures how consistently a model outputs the correct decision across repeated executions. Given the ground-truth decision $d^*$, decision stability is defined as $\hat{\theta} = p(d^* \mid \mathcal{M}, \mathcal{S}, s)$. This metric estimates the probability that a model selects the correct decision under stochastic decoding, and thus reflects robustness to execution noise.

### 5.2. Experimental Setup

We evaluate three representative models: Qwen3-VL-8B, UI-TARS-1.5-7B, and GUI-Owl-7B. For fair comparison, Qwen3-VL-8B-Instruct adopts the official sampling strategy of Qwen3-VL-8B-Thinking for cross-mode alignment, while all other models use their default inference settings.

Decision-level evaluation is performed on 3438 step tasks uniformly sampled from all benchmarks, with dataset statistics summarized in Table 17. For each model-step task pair $(\mathcal{M}, s)$ under a fixed sampling configuration $\mathcal{S}$, we conduct 8 rollout rounds with different random seeds, sampling 64 trajectories per round, for a total of 512 rollouts.

To isolate the effect of a single factor, we fix all other settings and estimate $p(d \mid \mathcal{M}, \mathcal{S}, s)$ under each condition. We then perform decision-level clustering and compare the resulting distributions using discrete decision diversity shift $\Delta_{\exp}\text{Div}$, stability shift $\Delta\hat{\theta}$, and baseline stability. We analyze two factors in turn: evaluation mode (Offline vs. SOEval) and inference mode (Instruct-only vs. Reasoning). Details of shift discretizations are provided in Appendix C.3.

### 5.3. How SOEval Improves Evaluation Contexts

We analyze the stability shift induced by SOEval relative to offline evaluation, denoted as $\Delta\hat{\theta}$ : Offline → SOEval, using the offline stability as the baseline for joint-distribution analysis. As shown in Figure 4, SOEval primarily increases stability for samples that are highly unstable under offline evaluation, while slightly suppressing stability for samples that are already stable.

| Model | increasing | decreasing |
|---|---|---|
| GUI-Owl-7B | (+0.4500) | (-0.3882) |
| UI-TARS-1.5-7B | (+0.2343) | (-0.2052) |
| Qwen3-VL-8B-Instruct | (+0.2004) | (-0.1503) |

*Table 4.* Stability shift mean on shift-trend categorized samples under SOEval, compared to the offline evaluation baseline.

We further examine how performance depends on the mix of online and offline interaction history. During SOEval rollouts, we record successful step-level outcomes, yielding a pool of on-policy step artifacts. At evaluation, history is con-

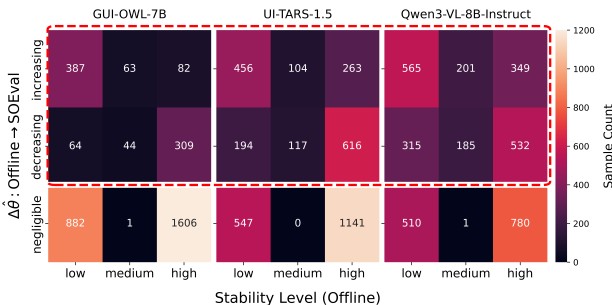

*Figure 4.* Joint discrete distribution between offline stability and SOEval-induced stability shifts. Mean stability shift of SOEval over the offline baseline by shift trend are summarized in Table 4.

structed by stochastically replacing offline steps with these on-policy artifacts, enabling controlled online–offline history mixing. By varying substitution probability over time, we define three regimes: increasing, stationary, and decreasing on-policy mixing strategies. We quantify online history injection using the online source ratio (OSR), the fraction of history offline steps replaced by on-policy artifacts, and measure its effect on stepwise exact match. We further characterize temporal bias via the sampling probability gap between the first and last history steps. Implementation details are summarized in Appendix D.2.1 and D.2.2.

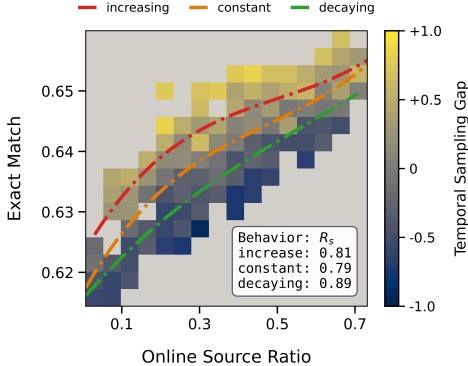

*Figure 5.* Relationship between exact match and online source ratio according to different on-policy mixing strategies, with Spearman correlation coefficients $R_s$ reported.

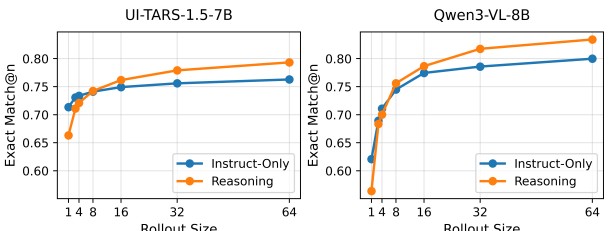

*Figure 6.* PASS@n performance according to rollout size $n$.

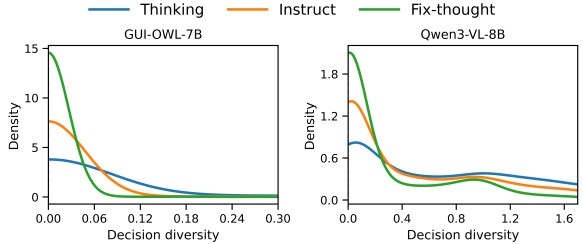

*Figure 7.* Diversity distribution under different inference modes.

We evaluate GUI-Owl-7B across temporal sampling gaps in $[-1, 1]$. As shown in Figure 5, exact match increases monotonically with OSR across all regimes, indicating that incorporating on-policy history consistently improves evaluation performance. At comparable OSR levels, increasing schedules outperform stationary ones, which in turn outperform decreasing schedules. This pattern suggests that near-decision context is more important to enhance model performance, while excessive reliance on on-policy context may interfere with already stable decisions, as shown in Figure 4 and further discussed in Appendix D.5. This trade-off highlights a key limitation of current models: they lack an adaptive mechanism to balance sensitivity to near-decision context with the preservation of stable decision behavior.

### 5.4. How Reasoning Shapes Decision Behavior

Tables 2 and 3 show a consistent performance drop when switching from instruct-only to thinking mode across models, which persists under SOEval, indicating that the degradation stems from reasoning behavior itself rather than off-policy context mismatch. To analyze this effect, Figure 7 compares decision diversity across inference modes. In addition to instruct-only and thinking, we introduce a fixed-thought setting (Appendix E) that fixes the reasoning trace from thinking mode while keeping execution stochastic, isolating the effect of explicit reasoning content from reasoning variability. The results show that explicit reasoning strongly conditions execution decisions. Compared to instruct-only inference, reasoning produces a broader and more dispersed distribution. Figure 8 further reveals a systematic trade-off: reasoning improves performance on steps that are unstable under the instruct-only baseline, but degrades performance on already stable ones by increasing execution variance. Under strict PASS@1 evaluation, the gains on unstable samples are outweighed by failures on stable ones, resulting in an overall performance drop.

This trade-off becomes favorable under more tolerant evaluation regimes. As shown in Figure 6, reasoning underperforms at PASS@1, its relative performance improves steadily as the number of sampled executions increases, eventually surpassing instruct-only inference at PASS@8. This rever-

sal indicates that reasoning expands the space of viable execution paths, even though these paths are sampled less consistently in PASS@1 evaluation. Overall, these results show that reasoning and instruct-only inference correspond to different operating points along the stability–diversity spectrum, and their relative effectiveness depends critically on the test-time tolerance of the evaluation regime.

#### 5.4.1. REASONING–EXECUTION CONSISTENCY ANALYSIS

Figure 7 shows that even under a fixed-thought setting, where the decision distribution is highly concentrated for most tasks, a small number of outliers remain. This indicates an inherent mismatch between reasoning traces and execution outcomes under stochastic sampling. We therefore ask: *to what extent does this mismatch affect performance?*

To measure reasoning–execution consistency robustly, we adopt a two-stage majority voting scheme. For each reasoning trace, a judge model performs $n$ rollouts and selects the most frequent execution. We then aggregate these model-level majority executions across multiple judges and determine whether the cross-model consensus matches the actual execution. To validate the reliability of this procedure, we construct a dedicated benchmark by manually curating $324$ reasoning–execution consistent and $324$ inconsistent step tasks with explicit reasoning traces and corresponding executions. We use GUI-Owl-32B, Qwen3-VL-32B-Thinking, and UI-Venus-Navi-72B as judges, each performing 32 rollouts per task. As shown in Table 5, the proposed approach achieves reliable performance in detecting reasoning–execution consistency, with two-stage voting consistently outperforming single-model voting. Additional validation details, including a balanced confusion matrix and Wilson confidence intervals, are provided in Appendix F.

| Model | Acc. | TPR | TNR |
|---|---|---|---|
| **Single-Model Voting** | | | |
| GUI-Owl-32B | 0.8796 | 0.8241 | 0.9352 |
| Qwen3-VL-32B | 0.8611 | 0.7685 | 0.9537 |
| UI-Venus-Navi-72B | 0.8056 | 0.7315 | 0.8796 |
| **Two-Stage Voting** | 0.8981 | 0.8426 | 0.9537 |

*Table 5.* Performance of reasoning-execution consistency detection. TPR and TNR denote true positive and true negative rates.

| Exact Match | R-E Consistent | R-E Inconsistent |
|---|---|---|
| True | 5531 | 456 |
| False | 1976 | 2037 |
| Match Ratio | 73.70 | 18.29 |

*Table 6.* Reasoning–execution (R-E) consistency analysis.

We further examine the relationship between reasoning-execution consistency and execution performance on GUI-

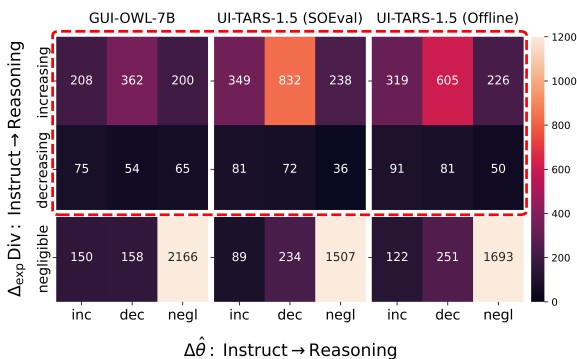

(a) Joint discrete distribution of stability and diversity shifts induced by reasoning relative to instruct-only mode.

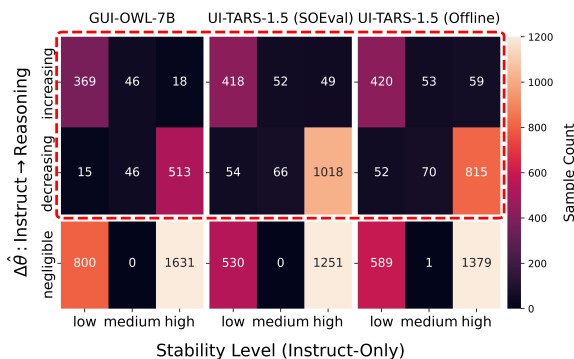

(b) Joint discrete distribution of instruct-only baseline stability and stability shifts introduced by reasoning.

*Figure 8.* Effect of reasoning on decision behavior. Left: reasoning consistently increases decision diversity for a substantial subset of tasks, but most diversity gains are accompanied by reduced decision stability. Right: Stability gains are concentrated on samples with low instruct-only baseline stability, while stability losses primarily occur on samples with high baseline stability.

Owl-7B. As shown in Table 6, reasoning–execution inconsistency is strongly associated with execution failure. The success rate is 73.7% for R-E consistent samples and 18.3% for R-E inconsistent samples, corresponding to an absolute difference of 55.4 percentage points. The same contingency table yields a relative risk of 4.03, an odds ratio of 12.50, and a Pearson $\chi^2 = 2389.58$ with $\phi = 0.489$ ($p \ll 10^{-3}$), indicating a strong statistical association despite potential noise in the consistency detector.

Among the sampled R-E inconsistent cases (Table 7), action-type mismatches constitute the largest error category, corresponding to failures in aligning the reasoning trace with the executed interaction type. Action-target mismatches form the second largest category and are primarily associated with grounding errors at the interaction target level. Action-type mismatches thus reflect higher-level decision errors, while action-target mismatches are more closely tied to perception and grounding. Qualitative examples illustrating these two failure modes are provided in Appendix F.

| Failure type | Count | Ratio | Interpretation |
|---|---|---|---|
| Action-type mismatch | 66 | 61.1% | Planning or action selection error |
| Action-target mismatch | 40 | 37.0% | Target grounding error |
| Invalid action | 2 | 1.9% | No valid effective interaction |

*Table 7.* Failure taxonomy over 108 sampled reasoning–execution inconsistent cases.

## 6. Related Work

Recent research has advanced along several complementary directions. Training-focused systems such as UItron (Zeng et al., 2025) emphasize large-scale data engineering, interactive infrastructure, and curriculum reinforcement learning, while UI-S1 (Lu et al., 2025b) explores semi-online reinforcement learning and introduces Semi-Online Performance (SOP), an evaluation metric designed to better align with interactive execution. In parallel, evaluation efforts such as MobiBench (Im et al., 2025) aim to improve offline assessment through mechanisms including multi-path awareness. Beyond training and evaluation, work on generalist agents and benchmarks such as AppAgent (Zhang et al., 2023), OS-ATLAS (Wu et al., 2024), and OmniACT (Kapoor et al., 2024) extends capabilities to smartphone interaction, cross-platform operating systems, and desktop and web automation. Our work complements these efforts by combining a mobile-focused dataset and toolkit with controlled SFT and RL scaling studies, semi-online evaluation, and decision-level analysis of reasoning and action behavior, providing a systematic empirical study of the training, evaluation, and behavioral properties of GUI agents.

## 7. Conclusion

We present `HyperTrack`, a large-scale mobile GUI dataset, alongside `GUIEvalKit`, a unified evaluation framework that supports both offline and semi-online assessment of GUI agents. Our results show that performance scales approximately log-linearly with training data size, with reinforcement-based finetuning providing consistent robustness gains over supervised finetuning, especially in out-of-domain settings. Through semi-online evaluation, we demonstrate that incorporating on-policy interaction history enhances evaluation fidelity and yields stronger correlations with online evaluation metrics. Using decision diversity and decision stability, we further reveal a fundamental stability–diversity trade-off: reasoning mode broadens the space of viable decisions but often reduces decision stability, leading to lower PASS@1 performance while benefiting more tolerant evaluation regimes such as online evaluation. Overall, these findings underscore the importance of jointly considering training scale, evaluation methods, and reasoning behaviors when developing and benchmarking GUI agents.

## Impact Statement

Mobile GUI agents have the potential to improve accessibility, reduce repetitive user effort, and enable productivity-oriented automation across a wide range of applications. At the same time, these capabilities may introduce risks, including spam generation, deceptive activities, unauthorized account actions, and attempts to circumvent user safeguards. To mitigate such risks, HyperTrack excludes applications involving highly sensitive information, uses dedicated test accounts and controlled environments for data collection, and applies privacy filtering and data cleaning procedures before release. Beyond task success, future deployment-oriented evaluations should additionally assess privacy preservation, appropriate refusal behavior for sensitive actions, robustness against deceptive or adversarial UI states, and safeguards against large-scale or policy-violating automation.

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

# A. `HyperTrack` Dataset

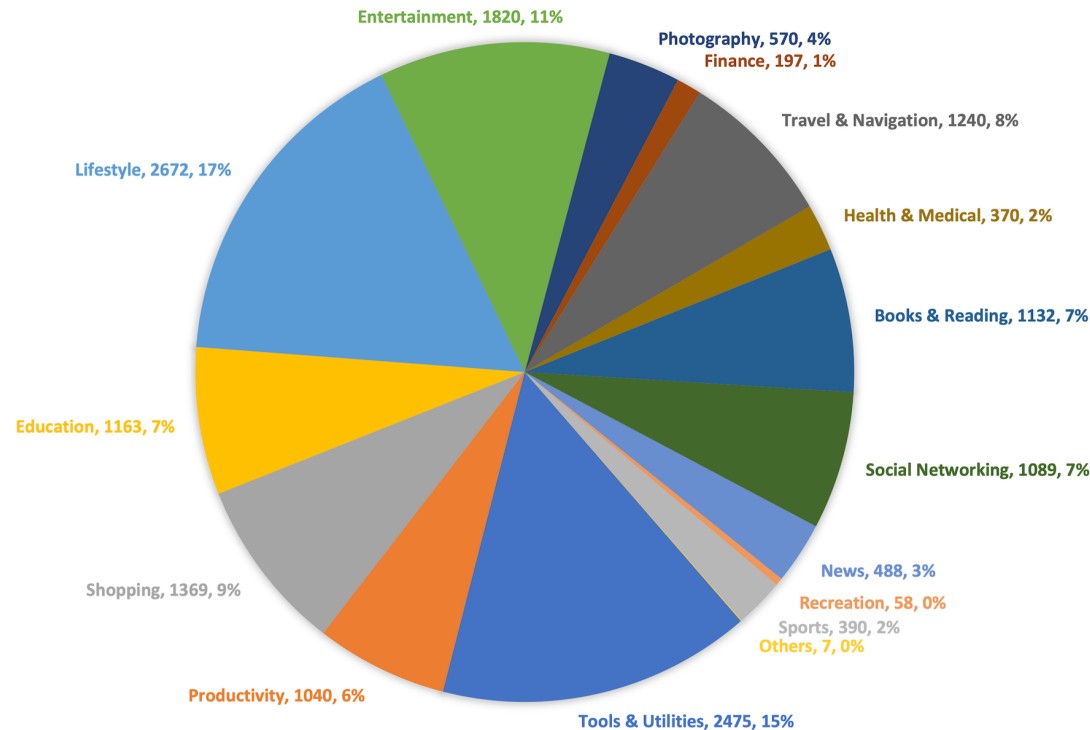

*Figure 9.* Distribution of app categories in the HyperTrack dataset.

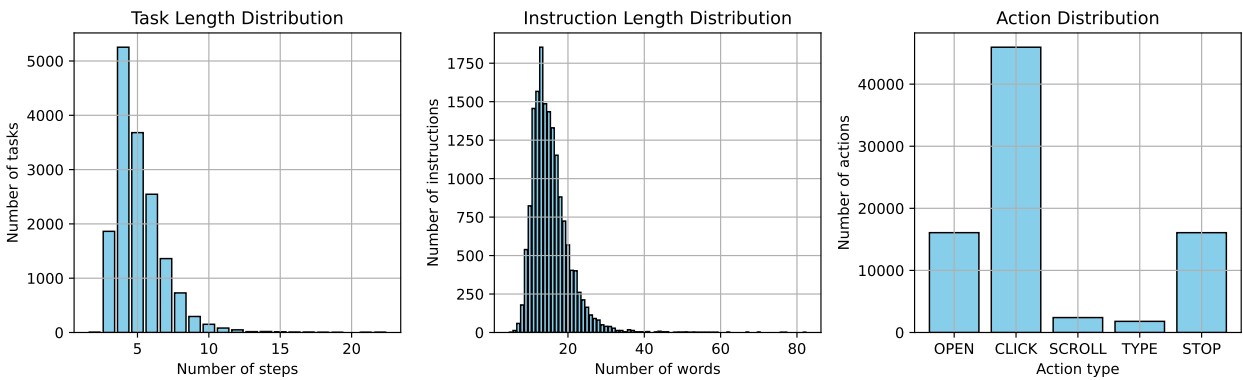

*Figure 10.* HyperTrack dataset statistics.

## A.1. Data Collection, Quality Control, and Privacy Constraints

The HyperTrack release format provides a static, reproducible representation of each interaction trajectory for offline training and evaluation. Each episode consists of a high-level user instruction, an ordered sequence of screenshots paired with textual screen observations, structured action annotations (including action type and natural-language action description), spatial parameters such as click coordinates and bounding boxes when applicable, and device- and episode-level metadata (e.g., device model, operating system, screen resolution, and episode identifier). This design enables reproducible evaluation without requiring access to live applications or redistribution of original app environments.

HyperTrack is constructed through a multi-stage annotation and filtering pipeline designed to ensure both scalability and consistency. The data collection process covers a diverse set of Chinese mobile applications and task scenarios. The high-level user instructions are constructed through a hybrid strategy, including manually written task descriptions as well as

| Split | Sub-splits | # Episodes | # Steps |
|---|---|---|---|
| Train | - | 13555 | 69223 |
| Val | - | 160 | 834 |
| Test | IDD | 562 | 2843 |
| | Unseen App | 664 | 3353 |
| | Unseen Device | 735 | 3929 |
| | Unseen App & Device | 404 | 2122 |

*Table 8.* Details on HyperTrack train, validation, and test splits.

template-based generation and expansion guided by application functionality trees and large language models. Annotators follow a structured annotation protocol that jointly models task intent, action semantics, screen state, and spatial grounding. A custom annotation tool supports structured action space specification and interactive bounding-box labeling, while a multi-round quality control process is applied to ensure task coherence, action correctness, and spatial annotation accuracy. Episodes failing quality checks are returned for revision or removed.

To enable safe release, all screenshots undergo a privacy filtering and sanitization process. Visual content is treated as the primary evidence, with textual observations used as supplementary signals. The filtering pipeline removes or masks personal identifiers, account credentials, financial and payment information, faces and avatars, private conversational content, precise location traces, device identifiers, and other sensitive attributes. When reliable anonymization cannot be guaranteed while preserving task-relevant GUI structure, the episode is excluded. A dataset entry is considered release-compliant only if no personally identifiable or sensitive information can be reconstructed, while preserving sufficient GUI semantics for downstream task understanding.

### A.2. Data availability

The experiments in this paper use the full HyperTrack dataset. At the time of submission, the public HyperTrack link provides a test subset intended for preview and data-format inspection. The full dataset remains under internal privacy, compliance, and redistribution review and is planned for public release after approval.

### A.3. Additional Scaling Results for Qwen3-VL-8B-Thinking

Figure 11 reports the complementary Qwen3-VL-8B-Thinking scaling results under SFT, binary RL, and Gaussian spatial reward.

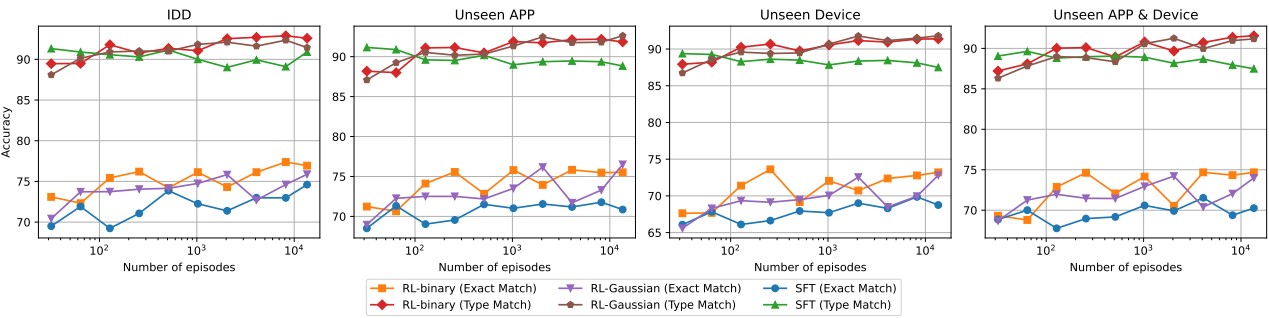

*Figure 11.* Qwen3-VL-8B-Thinking scaling results under SFT, binary RL, and Gaussian spatial reward.

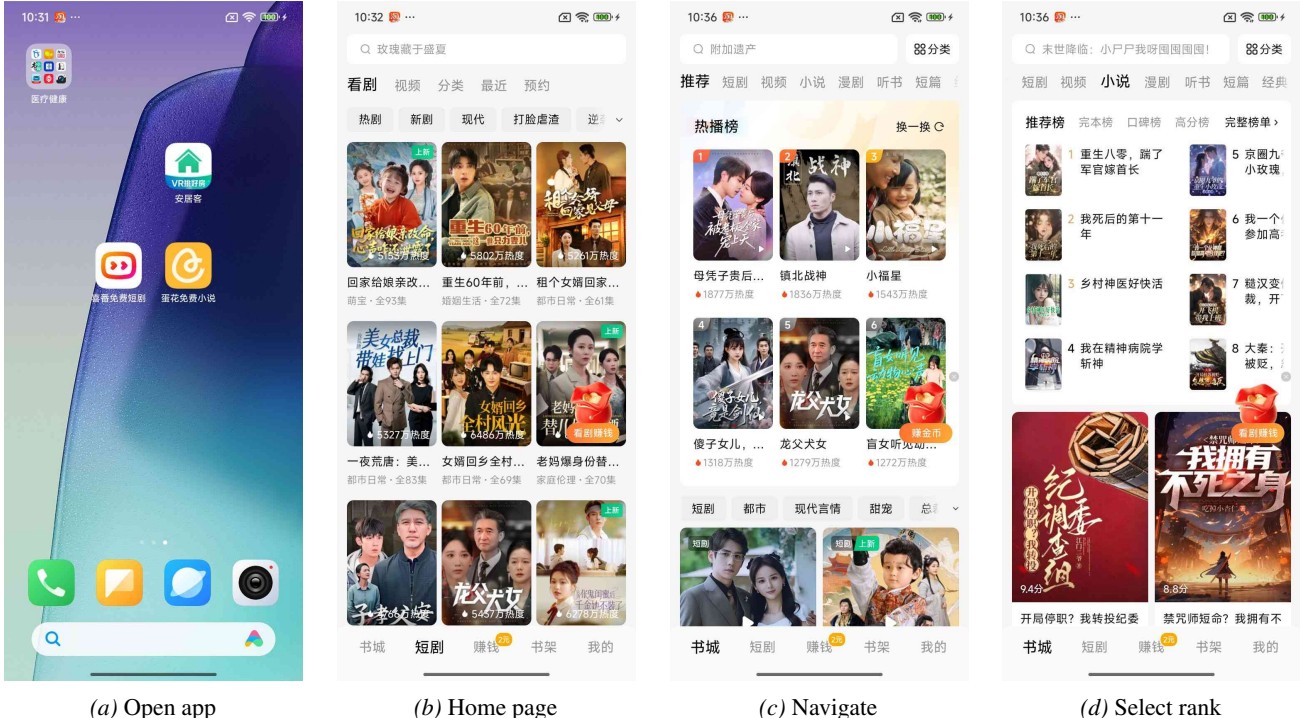

| *(a)* Open app | *(b)* Home page | *(c)* Navigate | *(d)* Select rank |

*Figure 12.* Representative privacy-cleared Chinese-language HyperTrack screenshots from one trajectory. The corresponding instruction is to view the reputation-ranking page in a Chinese novel app.

## B. `GUIEvalKit` Toolkit: Design Details

| Action | Description |
|---|---|
| CLICK(point = (x, y)) | Click at the relative coordinates (x, y) ∈ [0, 1000]. |
| LONG_PRESS(point = (x, y), duration = t) | Long press at point for duration t. |
| SCROLL(point = (x, y), to = direction) | Swipe from point in the specified direction. |
| TYPE(input = string) | Enter the given string into the focused text field. |
| OPEN(app = app_name) | Launch the application specified by app_name. |
| PRESS(press = button) | Press a system button (e.g., HOME, BACK, ENTER). |
| WAIT(duration = t) | Wait for duration t. |
| STOP(status = finish) | Terminate the task and mark it as finished. |

*Table 9.* The unified action space used in GUIEvalKit.

# C. Decision-Level Evaluation

## C.1. Clustering-Based Execution–Decision Abstraction

### C.1.1. CONCEPTUAL EXPLANATION

We treat $\mathcal{D}^{(s)}$ as a model-agnostic semantic reference space that specifies *what can be meaningfully decided* at a given step task $s$. While $\mathcal{D}^{(s)}$ defines the space of valid decisions, a particular model $\mathcal{M}$ determines which decisions are reachable and how probability mass is distributed over them through its execution behavior. Empirically, executions corresponding to the same semantic decision tend to form high-density modes in the execution space. These modes exhibit stable cluster cores across repeated rollouts, despite stochastic sampling and execution noise.

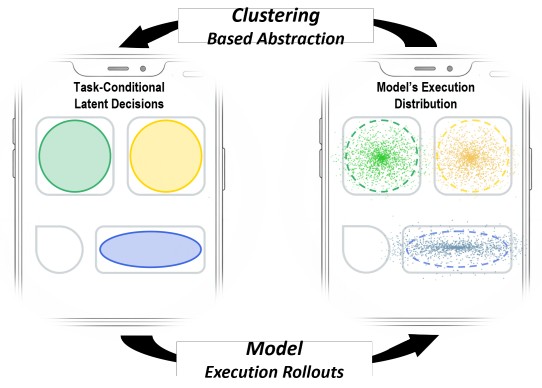

**Clustering as decision recovery.** We apply density-based clustering to the empirical execution samples of $\mathcal{M}$ for a given step task $s$, yielding a model-specific decision subspace $\mathcal{D}^{(s)}_{\mathcal{M}} \subseteq \mathcal{D}^{(s)}$. Each resulting cluster corresponds to a latent decision unit $d_k$, whose core approximates the canonical execution mode.

**Neighborhood scale as matching threshold.** The neighborhood scale (e.g., the $\varepsilon$-neighborhood radius for a CLICK action) jointly determines the clustering granularity and the induced matching criterion $\mathbf{1}(c(e_i) = d_k)$. As a result, the empirical histogram of cluster assignments directly defines the induced decision distribution $p(d_k)$, from which decision diversity is computed.

**Robustness to noise.** Low-density outliers correspond to sporadic or noisy executions. Their contribution to decision entropy is negligible since $\lim_{p \to 0} p \log p = 0$, ensuring that the resulting decision-level statistics are robust to execution noise.

Clustering-Based Execution–Decision Abstraction.

By visualizing execution distributions across multiple models and representative step tasks, we empirically observe the following patterns:

- High-density cluster cores consistently correspond to valid and actionable GUI affordances, rather than spurious or invalid executions.

- A subset of prominent density modes is shared across different models, providing empirical evidence for the existence of a model-agnostic decision space $\mathcal{D}^{(s)}$.

- Low-density noise is sparse and contributes negligibly to overall decision entropy.

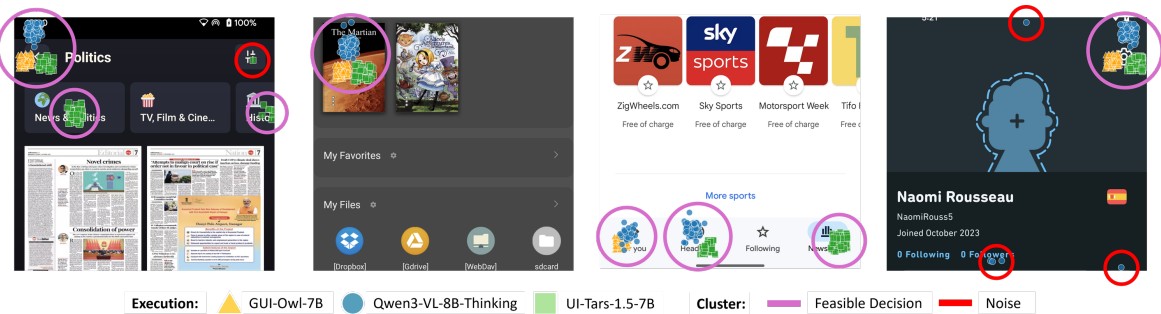

Execution distribution showcases across step tasks and models.

### C.1.2. IMPLEMENTATION DETAILS

We first partition all execution instances $\{e\}$ by action type under a unified action space:

$$\mathcal{A} = \{\texttt{CLICK}, \texttt{LONG\_PRESS}, \texttt{SCROLL}, \texttt{OPEN}, \texttt{TYPE}, \texttt{PRESS}, \texttt{WAIT}, \texttt{STOP}\}.$$

This results in a set of action-conditioned execution subsets $\{\mathcal{E}_a\}_{a \in \mathcal{A}}$, where $\mathcal{E}_a$ denotes executions of type $a$.

For each action type $a$, we independently apply density-based clustering to $\mathcal{E}_a$, producing cluster assignments $\{\mathcal{C}(e)\}_{e \in \mathcal{E}_a}$ and the corresponding clustered execution groups $\{\mathcal{E}_{a,c}\}_{\mathcal{C}}$. Each cluster $\mathcal{E}_{a,c}$ is treated as a latent decision unit $d_{a,c}$. Finally, we aggregate all clustered execution modes across action types to form the model-specific decision space $\mathcal{D}_{\mathcal{M}}^{(s)}$:

$$\mathcal{D}_{\mathcal{M}}^{(s)} = \bigcup_{a \in \mathcal{A}} \{d_{a,c}\}.$$

**CLICK & LONG_PRESS.** For spatial actions, we measure distances using the Euclidean $\ell_2$ norm over normalized screen coordinates, with $x, y \in [0, 1000]$. We apply DBSCAN with a neighborhood radius $\epsilon = 70$ to identify dense execution modes corresponding to distinct interaction targets.

**TYPE & OPEN.** For text-assignment actions, we cluster execution strings using a two-stage incremental matching scheme designed to balance precision and coverage. The first stage emphasizes high-precision matches based on containment relations, explicitly suppressing spurious matches caused by severe content truncation. The second stage allows controlled expansion using a stricter edit-distance criterion. Executions that satisfy neither condition are assigned to new clusters. The complete procedure is described in Algorithm 1.

---

**Algorithm 1** Two-Stage Clustering for Text-Assignment Executions

---

1: **Thresholds:** normalized edit distance ED with $\tau_{\text{loose}} = 0.3$ and $\tau_{\text{strict}} = 0.1$
2: **Input:** execution strings $X = \{x_1, \ldots, x_n\}$
3: **Output:** cluster set $\mathbb{C} = \{c_1, \ldots, c_m\}$ with prototypes $\Pi = \{\pi_1, \ldots, \pi_m\}$
4: Initialize $\Pi \leftarrow \{x_1\}$; $\mathbb{C} \leftarrow \{\{x_1\}\}$
5: **for** $i \leftarrow 2$ **to** $n$ **do**
6:     $x \leftarrow x_i$
7:     **if** $\exists \pi_j \in \Pi$ s.t. $\text{Contain}(x, \pi_j) \wedge \text{ED}(x, \pi_j) \leq \tau_{\text{loose}}$ **then**
8:         add $x$ to the corresponding cluster $c_j \in \mathbb{C}$
9:     **else**
10:         $\Gamma \leftarrow \{ j \mid \pi_j \in \Pi \wedge \text{ED}(x, \pi_j) \leq \tau_{\text{strict}} \}$
11:         **if** $\Gamma \neq \emptyset$ **then**
12:             $\varpi \leftarrow \arg \min_{j \in \Gamma} \text{ED}(x, \pi_j)$
13:             add $x$ to cluster $c_\varpi \in \mathbb{C}$
14:         **else**
15:             $m \leftarrow m + 1$
16:             create cluster $c_m \leftarrow \{x\}$ and set its prototype $\pi_m \leftarrow x$
17:             add $c_m$ to $C$ and $\pi_m$ to $\Pi$
18:         **end if**
19:     **end if**
20: **end for**

---

**SCROLL & PRESS.** For actions with discrete parameters — including SCROLL, parameterized by direction $d \in \{\text{up}, \text{down}, \text{left}, \text{right}\}$, and PRESS, parameterized by button type $b \in \{\text{BACK}, \text{HOME}, \text{ENTER}\}$ — the execution space is inherently finite and categorical. Clustering therefore reduces to grouping executions with identical literals, and clusters are fully characterized by their occurrence counts.

Formally, given a sequence of executions with associated literals $\{\ell_1, \ldots, \ell_n\}$, clustering amounts to aggregating identical literals, yielding the cluster

$$\{\ell_1, \ldots, \ell_n\} \longrightarrow \{ c_\ell \; : \; c_\ell = \{\ell_i \mid \ell_i = \ell\}, \; |c_\ell| = \text{count}(\ell) \}.$$

**WAIT & STOP.** For control actions, the action type alone fully specifies the execution. Consequently, all executions of WAIT and STOP are grouped solely by action type, each forming a single trivial cluster.

### C.1.3. CASE STUDY: STABILITY OF CURRENT CLUSTERING SCHEME

**CLICK & LONG_PRESS.**    To assess the stability of our clustering strategy for continuous execution points, we conduct a reachability analysis using OPTICS. The reachability plot provides a natural way to identify persistent density structures: peaks whose reachability distance exceeds a reference threshold ($\epsilon = 70$) are interpreted as indicating the emergence of a new cluster. Figures 13 and 14 illustrate the robustness of $\epsilon = 70$ under execution distributions that exhibit subtle geometric variations while corresponding to the same underlying click decision. In these cases, execution points remain concentrated on a single UI component, even though their spatial distribution may vary. When $\epsilon$ is reduced below 70, the distribution in Figure 14 is split into multiple clusters, resulting in an artificial increase in the estimated number of decisions. This behavior suggests that smaller neighborhood scales are overly sensitive to within-component micro-structures.

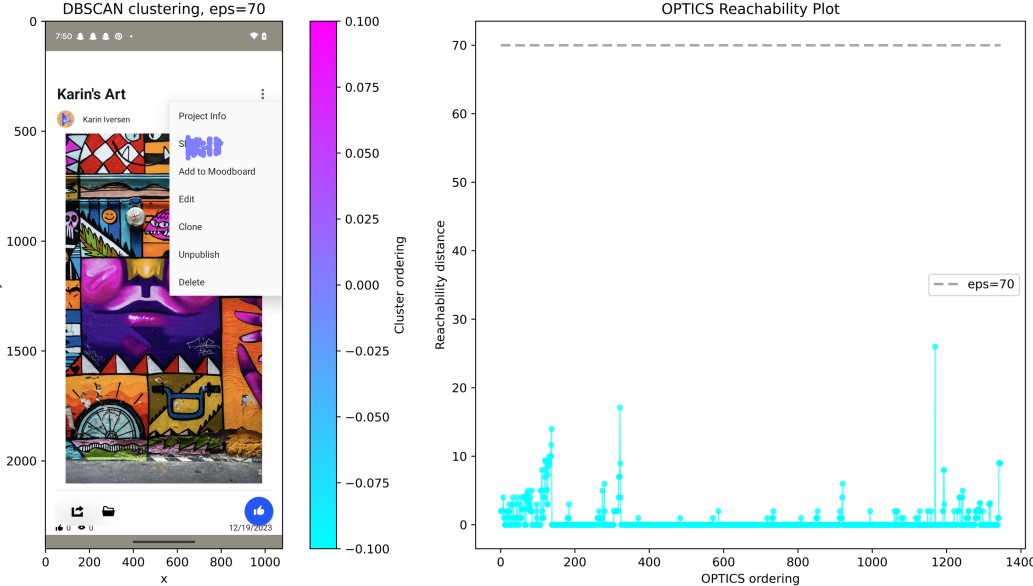

*Figure 13.* Click executions clustering. Execution points are relatively uniformly dispersed while remaining clearly concentrated on the target UI component, forming a single stable cluster.

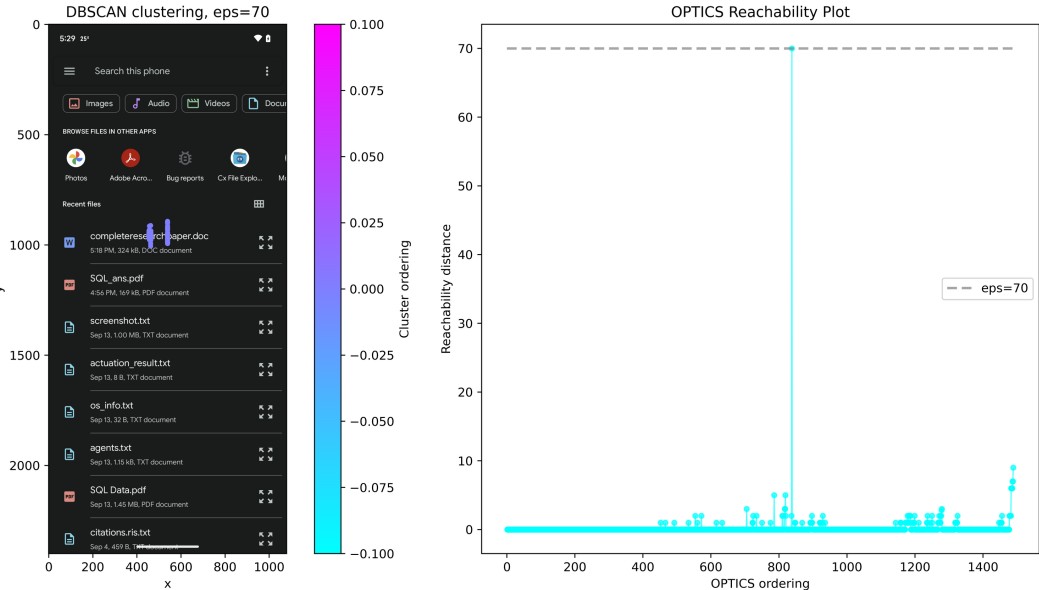

*Figure 14.* Click executions clustering. The execution distribution exhibits two well-separated but semantically aligned substructures. Due to the geometric shape of the target component, both subclusters are centered on the same UI element and correspond to a single decision.

Figure 15 presents a scenario in which two semantically distinct click decisions—targeting different UI elements—are cleanly separated into two well-defined clusters. In this case, $\epsilon = 70$ correctly recovers the underlying decision structure without ambiguity. Figure 16 highlights a more challenging configuration. Here, three major clusters correspond to distinct functional regions of the interface. However, due to a compact and dense page layout with multiple small, adjacent UI elements, execution points targeting different components in the top-left region are merged into a single cluster under $\epsilon = 70$, which leads to a mild underestimation of decision diversity.

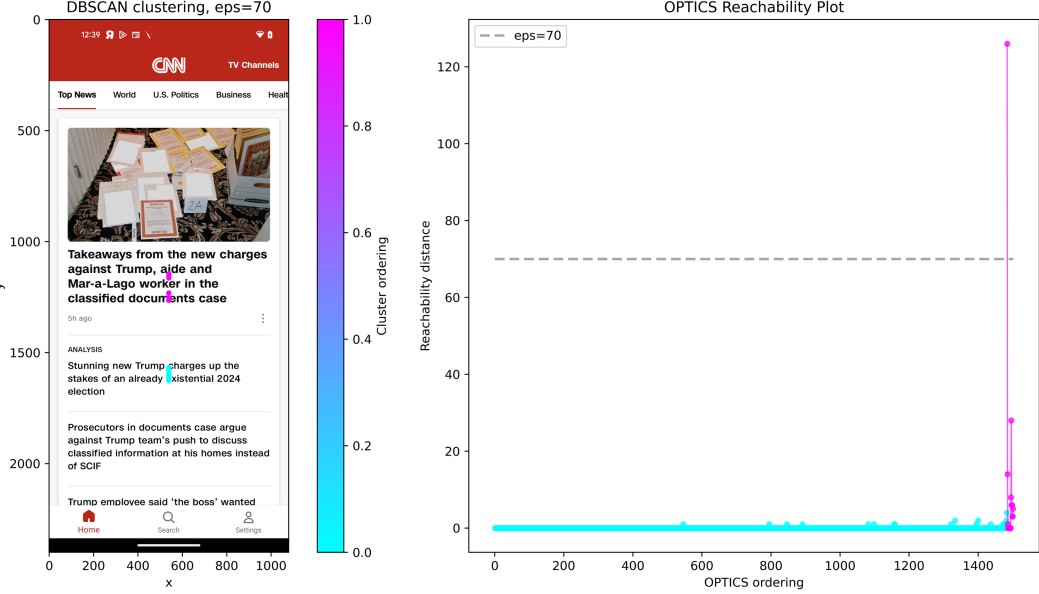

*Figure 15.* Click Executions Clustering. Execution points corresponding to clicks on different news items are clearly separated into two distinct clusters, reflecting two semantically different decisions.

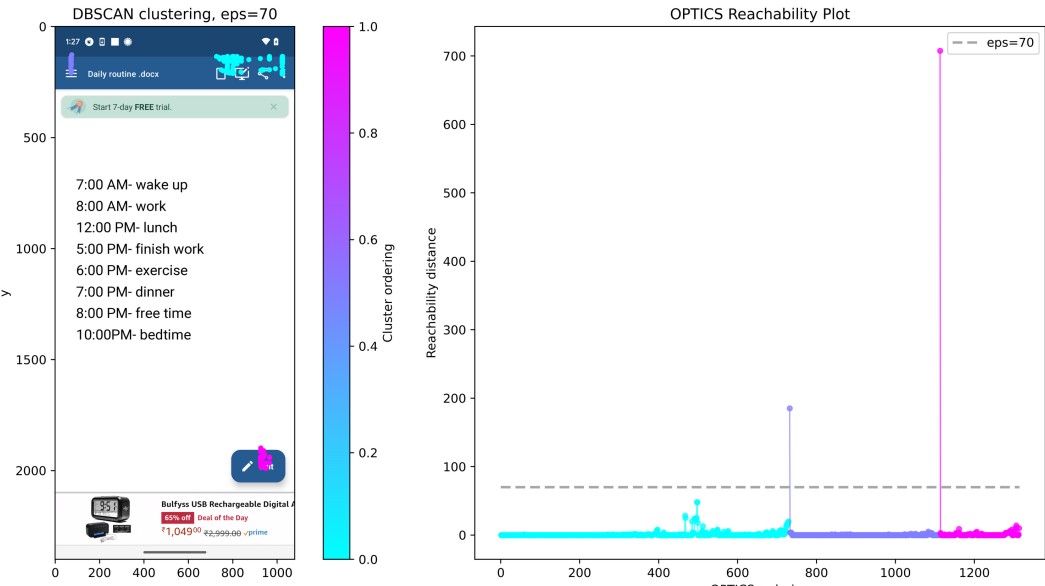

*Figure 16.* Click Executions Clustering. Three major clusters correspond to distinct high-level functional regions of the interface. However, under a compact and dense layout with multiple adjacent small UI elements (top-left region), execution points targeting different components are merged into a single cluster at $\epsilon = 70$, leading to a mild underestimation of decision diversity.

In practical GUI tasks, the majority of meaningful interactions occur on visually salient and well-separated UI components. Overly small neighborhood thresholds tend to fragment these dominant interaction patterns, systematically inflating decision diversity estimates. We therefore consider it preferable to sacrifice some fine-grained micro-structures, detectable only under very small $\epsilon$, in exchange for stability and robustness in the common case. Based on this trade-off, we empirically fix the neighborhood threshold for DBSCAN at $\epsilon = 70$.

### C.1.4. SENSITIVITY TO CLUSTERING CHOICES

We evaluate the robustness of the clustering-induced decision abstraction by varying the DBSCAN neighborhood radius and comparing $L_1$ and $L_2$ distance metrics. Although the absolute support size decreases as $\epsilon$ moves away from an extremely fine-grained regime, the supporting-size structure remains aligned across a broad range of thresholds. Table 10 reports the average support size and cross-$\epsilon$ supporting-size correlations.

| $\epsilon$ | 5 | 15 | 35 | 50 | 70 | 100 |
|---|---|---|---|---|---|---|
| Avg. $|\mathcal{D}_\epsilon|$ | 8.34 | 4.24 | 3.33 | 3.14 | 2.98 | 2.84 |

*Table 10.* Average decision-support size under different DBSCAN neighborhood thresholds.

| $\epsilon$ pair | 5–70 | 15–70 | 25–70 | 35–70 | 70–100 | 35–100 |
|---|---|---|---|---|---|---|
| Supporting-size $R^2$ | 0.44 | 0.87 | 0.95 | 0.98 | 0.99 | 0.96 |

*Table 11.* Cross-sample supporting-size alignment across clustering thresholds.

For spatial actions, all coordinates are normalized to $\Omega = [0, 1000]^2$. For a fixed action type, two clustered decision distributions induce empirical measures $\mu_a = \sum_i p_i \delta_{x_i}$ and $\nu_a = \sum_j q_j \delta_{y_j}$. We compute the 1-Wasserstein distance with Euclidean ground cost and report the normalized value

$$W_1^{\text{norm}}(\mu_a, \nu_a) = \frac{W_1(\mu_a, \nu_a)}{1000\sqrt{2}},$$

where $1000\sqrt{2}$ is the diameter of the normalized screen domain. This measure captures whether changing $\epsilon$ causes local refinement/coarsening or a large spatial reorganization of decision supports. As shown in Table 12, the distances remain low and stable across the tested range.

| $\epsilon$ pair | 5→30 | 10→30 | 30→50 | 50→70 | 70→100 | 100→140 |
|---|---|---|---|---|---|---|
| Avg. $W_1^{\mathrm{norm}}$ | 0.163 | 0.162 | 0.163 | 0.163 | 0.162 | 0.162 |

*Table 12.* Normalized Wasserstein distance between $\epsilon$-induced spatial decision distributions.

We also compare $L_1$ and $L_2$ clustering after scale adjustment $\hat{\epsilon}_2 = \frac{\sqrt{2}+1}{2}\epsilon_1$. The resulting decision distributions are nearly identical, with cross-metric normalized Wasserstein distances below $5 \times 10^{-4}$ in Table 13.

| $\epsilon_1$ vs. $\hat{\epsilon}_2$ | 10 | 20 | 30 | 50 | 70 | 100 |
|---|---|---|---|---|---|---|
| Avg. $W_1^{\mathrm{norm}}$ | $4.14{\times}10^{-4}$ | $2.70{\times}10^{-4}$ | $2.46{\times}10^{-4}$ | $2.11{\times}10^{-4}$ | $3.56{\times}10^{-4}$ | $4.12{\times}10^{-4}$ |

*Table 13.* Cross-metric normalized Wasserstein distance between $L_1$ and scale-adjusted $L_2$ clustering.

Most importantly, the behavioral conclusions remain stable under these variations. Tables 14 and 15 show that reasoning still predominantly increases diversity, while many diversity gains coincide with reduced stability. Thus, $\epsilon$ should be interpreted as a geometric prior over action equivalence rather than a freely tuned outcome variable; smaller values favor fine-grained partitions, while larger values favor more abstract decision merging.

| Setting | Diversity | GUI-Owl-7B $\hat{\theta}\uparrow$ | GUI-Owl-7B $\hat{\theta}\downarrow$ | UI-TARS-1.5 $\hat{\theta}\uparrow$ | UI-TARS-1.5 $\hat{\theta}\downarrow$ |
|---|---|---|---|---|---|
| Baseline $\epsilon = 70$ | Div ↑ | 208 | 362 | 349 | 832 |
| Baseline $\epsilon = 70$ | Div ↓ | 75 | 54 | 81 | 72 |
| $\epsilon = 30$ | Div ↑ | 215 (+7) | 362 (+0) | 362 (+13) | 842 (+10) |
| $\epsilon = 30$ | Div ↓ | 77 (+2) | 59 (+4) | 81 (+0) | 78 (+6) |
| $\epsilon = 90$ | Div ↑ | 207 (-1) | 360 (-2) | 344 (-5) | 826 (-6) |
| $\epsilon = 90$ | Div ↓ | 75 (+0) | 52 (-2) | 81 (+0) | 73 (+1) |
| $\epsilon = 140$ | Div ↑ | 205 (-3) | 357 (-5) | 336 (-13) | 804 (-28) |
| $\epsilon = 140$ | Div ↓ | 72 (-3) | 52 (-2) | 77 (-4) | 73 (+1) |

*Table 14.* Sensitivity of the diversity–stability trend to the DBSCAN neighborhood threshold under the $L_2$ metric. Parentheses report differences from the $\epsilon = 70$ baseline.

| Setting | Diversity | GUI-Owl-7B $\hat{\theta}\uparrow$ | GUI-Owl-7B $\hat{\theta}\downarrow$ | UI-TARS-1.5 $\hat{\theta}\uparrow$ | UI-TARS-1.5 $\hat{\theta}\downarrow$ |
|---|---|---|---|---|---|
| $\hat{\epsilon}_2 = f(\epsilon_1 = 70)$ | Div ↑ | 207 (-1) | 362 (+0) | 345 (-4) | 830 (-2) |
| $\hat{\epsilon}_2 = f(\epsilon_1 = 70)$ | Div ↓ | 75 (+0) | 53 (-1) | 82 (+1) | 72 (+0) |

*Table 15.* Robustness of the diversity–stability trend under $L_1$ versus scale-adjusted $L_2$ clustering. Parentheses report differences from the $L_2$, $\epsilon = 70$ baseline.

**TYPE & OPEN.**  We report representative clustering results for text-based executions in Table 16, produced by the two-stage text clustering procedure described earlier. The results demonstrate that the method effectively separates executions conveying substantively different information, while remaining robust to truncation, stylistic variation, and incomplete inputs.

| Text Cluster Prototype | Count |
|---|---|
| There will be a Science Fair **in our city next month**. | 225 |
| There will be a Science Fair **at Lincoln High School tomorrow**. | 155 |
| There will be a great science fair event **this Sunday**. | 114 |
| Science Fair will come soon | 8 |
| There will be a Science Fair Event **next week**. | 7 |
| There will be a science fair held **in our school tomorrow** | 3 |

*(a)* Event announcement texts

| Text Cluster Prototype | Count |
|---|---|
| m so **happy** today | 151 |
| m so happy today **because my family came together** | 139 |
| m | 80 |
| I'm **tired** | 47 |
| I am really happy today **because it's my birthday!** | 12 |
| I am **alone** | 3 |

*(b)* Emotion expression texts

*Table 16.* Text Executions Clustering. Representative clustering results for text-based executions. Each subtable reports a cluster prototype and the number of executions assigned to it.

## C.2. Rollouts for Execution Distribution

To estimate the decision distribution at each step task $s$, we perform $n$ independent rollouts $\{e_j\}_{j=1}^n$ serving as a Monte Carlo approximation of the model $\mathcal{M}$'s implicit execution distribution under a fixed sampling strategy $\mathcal{S}$:

$$e_j \sim P(\cdot \mid \mathcal{M}, \mathcal{S}, s).$$

We choose the rollout size $n$ to be sufficiently large to provide a reliable approximation of the distribution for each step, while also allowing coverage across a diverse set of tasks within a reasonable time budget. To this end, we uniformly sample an experimental subset from the benchmarks supported in GUIEvalKit. The statistics of this subset are summarized in Table 17.

| Benchmark | #Episodes | #Step Tasks | Avg. Episode Length |
|---|---|---|---|
| AndroidControl | 101 | 763 | 7.55 |
| AiTZ | 73 | 724 | 9.92 |
| GUI Odyssey | 44 | 633 | 14.39 |
| CAGUI Agent | 85 | 677 | 7.96 |
| HyperTrack | 116 | 641 | 5.53 |
| Total | 419 | 3438 | 8.21 |

*Table 17.* Experiment subset statistics across supported benchmarks.

| Metrics | Qwen3-VL-8B-Thinking |
|---|---|
| Exact Match | 87.75 |
| Episode Success | 33.65 |

| Metrics | GUI-Owl-7B |
|---|---|
| Exact Match | 82.02 |
| Episode Success | 23.63 |

*Table 18.* Aggregate performance on the constructed evaluation subset.

For each step task in the subset, we perform 8 rollout rounds across 8 machines, each equipped with $8 \times$ Nvidia H100 GPUs, with each round producing 64 rollouts. In addition, these rollouts yield a substantial number of on-policy correct executions for the experimental subset. This enables direct reuse of the collected rollouts for subsequent sensitivity analysis under SOEval, particularly for controlled history formulation. Table 18 reports statistics of the on-policy correct results available on this subset.

### C.2.1. RANDOM SEEDS

We use a fixed set of random seeds across rounds to encourage independence while allowing rollouts from different rounds to be safely aggregated. The seeds used in our experiments include:

$$7278727,\ 7779397,\ 7771087,\ 7867747,\ 7977857,\ 5113051,\ 9581717,\ 20000303, \ldots$$

### C.2.2. EPISODE-LEVEL PROGRESS ACROSS SEEDS

To complement the step-level decision analysis, we report episode-level progress over eight seeds on the same decision-level evaluation subset. These subset results are not a replacement for full multi-seed evaluation over the entire benchmark suite, but they quantify the seed-level variation of the rollout set used in our behavioral analyses.

| Model | S1 | S2 | S3 | S4 | S5 | S6 | S7 | S8 | Mean | 95% CI |
|---|---|---|---|---|---|---|---|---|---|---|
| Qwen3-VL-8B-Thinking | 0.1090 | 0.1029 | 0.1072 | 0.1124 | 0.1162 | 0.1162 | 0.1213 | 0.1248 | 0.1138 | [0.1076, 0.1199] |
| GUI-Owl-7B | 0.1892 | 0.1932 | 0.1858 | 0.1916 | 0.1868 | 0.1968 | 0.1898 | 0.1883 | 0.1902 | [0.1872, 0.1932] |
| UI-TARS-1.5 | 0.1761 | 0.1740 | 0.1759 | 0.1685 | 0.1670 | 0.1849 | 0.1834 | 0.1835 | 0.1767 | [0.1709, 0.1824] |

*Table 19.* Episode-level progress across eight random seeds on the decision-level evaluation subset.

### C.2.3. MODEL CONFIGURATION.

All models are evaluated using their officially recommended prompts for mobile GUI tasks.

**GUI-Owl-7B.**

```
SamplingParams(
  max_tokens=2048,
  temperature=0.1,
  top_p=0.001,
  top_k=1,
  repetition_penalty=1.05,
  n=1,
  stop_token_ids=[]
)

Instruction_Template = '''The user query:
{instruction}
Task progress (You have done the following operation on the current device): Step 1: {Action}; ...; Step n: {Action
    };.
Before answering, explain your reasoning step-by-step in <thinking></thinking> tags, and insert them before the <
    tool_call></tool_call> XML tags.
After answering, summarize your action in <conclusion></conclusion> tags, and insert them after the <tool_call></
    tool_call> XML tags.
'''
```

**Qwen3-VL-8B.**

For cross-mode alignment, `Qwen3-VL-8B-Instruct` uses `THINKING_SAMPLING_PARAMS` in our experiments.

```
INSTRUCT_SAMPLING_PARAMS = SamplingParams(
  top_p=0.8,
  top_k=20,
  temperature=0.7,
  repetition_penalty=1.0,
  presence_penalty=1.5,
  max_tokens=2048,
  n=1
)
THINKING_SAMPLING_PARAMS = SamplingParams(
  top_p=0.95,
  top_k=20,
  temperature=1.0,
  repetition_penalty=1.0,
  presence_penalty=0.0,
  max_tokens=2048,
  n=1
)

Instruction_Template = '''{instruction}'''
```

**UI-Tars-1.5-7B.**

```
SamplingParams(
  max_tokens=2048,
  temperature=0.1
)
```

For UI-Tars-1.5-7B, we additionally implement a prompt-level switch to enable or disable explicit reasoning.

```
Instruct_Template=You are a GUI agent. You are given a task and your action history, with screenshots. You need to
    perform the next action to complete the task.
## Output Format
```
{% if enable_think -%}
Thought: ...
{% endif -%}
```

```
Action: ...
```
## Action Space

click(start_box='<|box_start|>(x1,y1)<|box_end|>')
long_press(start_box='<|box_start|>(x1,y1)<|box_end|>')
type(content='') #If you want to submit your input, use "\n" at the end of `content`.
scroll(start_box='<|box_start|>(x1,y1)<|box_end|>', direction='down or up or right or left')
open_app(app_name='')
drag(start_box='<|box_start|>(x1,y1)<|box_end|>', end_box='<|box_start|>(x3,y3)<|box_end|>')
press_home()
press_back()
finished(content='xxx') # Use escape characters \', \", and \n in content part to ensure we can parse the content in
    normal python string format.

{% if enable_think %}
## Note
- Use {% if language is none%}English{% else %}{{language}}{% endif %} in `Thought` part.
- Write a small plan and finally summarize your next action (with its target element) in one sentence in `Thought`
    part.

{% endif -%}
## User Instruction
{{instruction}}
```

### C.2.4. ROLLOUT EXAMPLE

We provide an example of GUI-Owl-7B rollout outputs for a single-step task below, illustrating the structure of model responses, predicted actions, and evaluation results collected during execution.

```
{
    "instruction": "Search for hotels in London",
    "result_samples": [
        {
            "response": "<thinking>\nI need to continue searching for hotels in London. ...",
            "answer": "{\"name\": \"mobile_for_gui_owl\", \"arguments\": {\"action\": \"system_button\",
→   \"button\": \"Enter\"}}",
            "thought": "I need to continue searching for hotels in London. The search bar is already filled with
→   the relevant query, indicating that I'm ready to execute the search command again if
→   necessary.\nAction: Press 'Enter' to initiate the search for hotels in London.",
            "conclusion": "press enter",
            "action": "PRESS",
            "prediction": {
                "PRESS": "ENTER"
            },
            "evaluation": {
                "type_match": false,
                "exact_match": false,
                ...
            }
        },
        {
            "response": "<thinking>\nI am currently viewing search results related to hotels in London. ...",
            "answer": "{\"name\": \"mobile_for_gui_owl\", \"arguments\": {\"action\": \"swipe\", \"coordinate\":
→   [259, 499], \"coordinate2\": [267, 239]}}",
            "thought": "I am currently viewing search results related to hotels in London. To proceed further, I
→   should explore these results to find suitable hotel options. Scrolling down will allow me to view
→   additional listings that might be relevant to my search criteria.",
            "conclusion": "Scroll down to view more hotel options in the search results.",
            "action": "SCROLL",
            "prediction": {
                "POINT": [
                    616,
                    685
                ],
                "to": "up"
            },
            "evaluation": {
                "type_match": true,
                "exact_match": true,
                ...
            }
        },
        {
            "response": "<thinking>\nI need to continue searching for hotels in London. ...",
            "answer": "{\"name\": \"mobile_for_gui_owl\", \"arguments\": {\"action\": \"click\", \"coordinate\":
→   [259, 154]}}",
            "thought": "I need to continue searching for hotels in London. The search bar is already filled with
→   the relevant query, so I should proceed to execute the search command to view the results.",
```

```
                    "conclusion": "Click on the search icon next to the search bar to initiate the search.",
                    "action": "CLICK",
                    "prediction": {
                        "POINT": [
                            616,
                            211
                        ]
                    },
                    "evaluation": {
                        "type_match": false,
                        "exact_match": false,
                        ...
                    }
                },
                ...
            ]
}
```

## C.3. Discretization of Decision-Level Metric Distribution

### C.3.1. DECISION DIVERSITY SHIFT DISCRETIZATION

To facilitate interpretable comparison of diversity shifts, we discretize entropy-based diversity through its corresponding effective support size. We reserve $\Delta$ for aligned semantics, and use $\Delta_{\text{exp}}$ to denote differences measured after exponentiation.

**Effective Support Size.**   Consider a discrete uniform distribution over $m$ states. Its entropy is

$$p_i = \frac{1}{m}, \qquad H_m = -\sum_{i=1}^{m} p_i \log p_i = \log m.$$

This establishes a natural monotonic mapping from entropy to an equivalent support size,

$$U : H \mapsto m, \qquad U(H) = \exp(H),$$

which admits a direct interpretation as the number of effectively activated states.

We therefore quantify decision diversity via the effective support size

$$N_{\text{eff}} \triangleq U(\text{Div}) = \exp(\text{Div}),$$

and compare diversity shifts using the exponentiated difference

$$\Delta_{\text{exp}}\text{Div} \triangleq \exp(\text{Div}_{\text{after}}) - \exp(\text{Div}_{\text{before}}).$$

**Thresholding of Diversity Shifts.**   We categorize changes in decision dispersion based on $\Delta_{\text{exp}}\text{Div}$ as follows:

$$\begin{cases} |\Delta_{\text{exp}}\text{Div}| \leq 0.1, & \text{negl / negligible change,} \\ \Delta_{\text{exp}}\text{Div} > 0.1, & \text{inc / increasing / expansion,} \\ \Delta_{\text{exp}}\text{Div} < -0.1, & \text{dec / decreasing / contraction.} \end{cases}$$

These thresholds correspond to fractional changes in effective support size and are invariant to the absolute scale of entropy, making them suitable for cross-task comparison.

**Limitations of Direct Entropy Differences.**   A natural alternative is to compare diversity shifts directly via $\Delta\text{Div}$. While entropy is well-suited for quantifying absolute dispersion, its finite differences are less appropriate for direct behavioral comparison, as they conflate multiple heterogeneous factors.

To make this explicit, we consider a dominant-mode perturbation regime, which is empirically satisfied in our experiments, where the most frequent decision cluster accounts for the majority of probability mass. Let the decision distribution be

$$p = \left(1 - \varepsilon, \ \varepsilon q_1, \ \ldots, \ \varepsilon q_m\right), \qquad \sum_{i=1}^{m} q_i = 1,$$

where $1 - \varepsilon$ denotes the dominant decision and $\varepsilon \ll 1$ represents a small perturbation mass distributed over secondary modes. The entropy decomposes as

$$H(p) = h(\varepsilon) + \varepsilon H(q), \qquad h(\varepsilon) = -(1 - \varepsilon) \log(1 - \varepsilon) - \varepsilon \log \varepsilon.$$

Under a local perturbation, the first-order variation of entropy satisfies

$$\delta H(p) \approx \log \frac{1 - \varepsilon}{\varepsilon} \, \delta\varepsilon \; + \; H(q) \, \delta\varepsilon \; + \; \varepsilon \, \delta H(q),$$

where $\delta(\cdot)$ denotes infinitesimal variations.

This expression shows that entropy change simultaneously couples perturbation magnitude, tail-mass allocation, and the internal structure of secondary decisions. As these components reside on incommensurate scales, $\Delta \mathrm{Div}$ does not admit a canonical or directly interpretable decomposition.

**Adopted Perspective.**   Rather than over-interpreting absolute entropy differences, we adopt $\Delta_{\mathrm{exp}}\mathrm{Div}$ as the primary measure for behavioral comparison. This choice preserves the monotonic relationship to entropy while yielding a linear-scale quantity that directly reflects expansion or contraction of the activated decision space, without introducing additional assumptions or auxiliary metrics.

### C.3.2. DECISION STABILITY DISCRETIZATION

Decision stability $\hat{\theta}$ is a linear metric in the sense that execution-level exact match is measured as an empirical mean of Bernoulli outcomes:

$$\widehat{\mathrm{EM}} \; = \; \mathbb{E}[\hat{\theta}].$$

Therefore, discretization for $\hat{\theta}$ and its shift is straightforward.

**Shift Discretization.**   Let $\Delta\hat{\theta}$ denote the stability shift between two evaluation contexts.

$$\begin{cases} |\Delta\hat{\theta}| = 0, & \text{negl / negligible change,} \\ \Delta\hat{\theta} > 0, & \text{inc / increasing / expansion,} \\ \Delta\hat{\theta} < 0, & \text{dec / decreasing / contraction.} \end{cases}$$

**Level Discretization.**   We set the high-stability threshold to $0.8$, which is stringent enough that even PASS@64 under multiple inference modes often falls short of it. We set the low-stability threshold to $0.4$, which approximately matches the lower bound of the exact-match rates observed in our offline evaluations.

$$\begin{cases} \hat{\theta} \leq 0.4, & \text{low,} \\ \hat{\theta} > 0.8, & \text{high,} \\ 0.4 < \hat{\theta} \leq 0.8, & \text{medium.} \end{cases}$$

## D. Semi-Online Evaluation

### D.1. Correlation with AndroidWorld Online Success

Table 20 provides the raw six-model data used in Figure 3. AW online success is treated as the online reference axis, while SOEval and offline metrics are computed from the static benchmark protocol. SOEval step exact match has the strongest reported linear association with AW online success among the compared metrics, while SOEval episode progress provides complementary but more episode-level evidence.

| Model | AW Online | SOEval Step EM | Offline Step EM | SOEval Progress |
|---|---|---|---|---|
| Qwen2.5-VL-3B-Instruct | 13.0 | 55.93 | 56.68 | 8.19 |
| Qwen3-VL-8B-Instruct | 47.6 | 62.84 | 60.39 | 9.00 |
| Qwen3-VL-4B-Thinking | 52.0 | 57.66 | 54.52 | 8.40 |
| UI-Venus-Navi-72B | 65.9 | 76.16 | 74.09 | 15.84 |
| GUI-Owl-7B | 66.4 | 67.37 | 65.49 | 12.01 |
| GUI-Owl-32B | 67.0 | 71.66 | 70.95 | 14.17 |
| $R^2$ | – | 0.6241 | 0.4821 | 0.5377 |
| Spearman $\rho$ | – | 0.7714 | 0.6571 | 0.7714 |

*Table 20.* Raw data for the SOEval/offline correlation with AndroidWorld online success. EM denotes step exact match.

## D.2. Sensitivity Analysis

### D.2.1. ONLINE SOURCE RATIO.

We quantify the overall degree of online history injection using the online source ratio (OSR):

$$\text{OSR} = \frac{\sum_{e,t} \mathbf{1}(\hat{a}_t = a_t)}{\sum_{e,t} 1}, \tag{8}$$

computed over all evaluation episodes $e$ and steps $t$ under fixed $(\mathcal{M}, s, \mathcal{S})$. This allows us to analyze how model performance metrics, such as stepwise exact match, respond to varying degrees of online history injection.

### D.2.2. HISTORY SELECTION STRATEGY

For the history simulation at step $i$ defined as

$$(\psi(a_0, o_0, \hat{a}_0, \hat{\tau}_0), \ldots, \psi(a_{i-1}, o_{i-1}, \hat{a}_{i-1}, \hat{\tau}_{i-1})), \tag{9}$$

where $\hat{\tau}$ captures the model's reasoning process, we define the history selection operator $\psi(\cdot)$ as

$$\psi(o_t, a_t, \hat{a}_t, \hat{\tau}_t) = \begin{cases} \varphi(o_t, \hat{a}_t, \hat{\tau}_t), & \text{if } \hat{a}_t = a_t \\ \phi(o_t, a_t), & \text{otherwise.} \end{cases} \tag{10}$$

This formulation allows us to explicitly control, at each time step, whether the history is constructed from an on-policy execution and its associated reasoning trace, or from the offline reference execution. By doing so, we can simulate a model's temporal behavior as it fluctuates between on-track and off-track decision states.

Thus we model the history mixing process as a Bernoulli sequence $I_t \sim \text{Bernoulli}(p_t)$, $I_t \in \{0, 1\}$ with a time-varying sampling probability $p_t$. By controlling the temporal evolution of $p_t$, we induce different regimes of history mixing that correspond to increasing, decreasing, or stationary on-track consistency over time.

To make these temporal regimes concrete and reproducible in evaluation, we instantiate $p_t$ using a normalized logistic schedule parameterized by the relative step position within an episode. For the $t$-th step in a history sequence of length $T$, we define the step ratio as $sr(t) = \frac{t+1}{T}$. Denoting the logistic sigmoid function as $\sigma(x) = (1 + \exp(-x))^{-1}$, we define the normalized logistic functions as

$$\text{nlogi}_+(x) = \frac{\sigma(\kappa(x - \mu)) - \sigma(-\kappa\mu)}{\sigma(\kappa(1 - \mu)) - \sigma(-\kappa\mu)}, \tag{11}$$
$$\text{nlogi}_-(x) = 1 - \text{nlogi}_+(x),$$

which map $x \in [0, 1]$ to $[0, 1]$ while preserving monotonicity. The normalization ensures that the minimum and maximum values of the schedule are independent of the shape parameters. Using these functions, we parameterize the sampling probability as

$$p_+(t) = p_{lb} + gap \cdot \text{nlogi}_+(sr(t)), \tag{12}$$
$$p_-(t) = p_{lb} + gap \cdot \text{nlogi}_-(sr(t)),$$

where $p_{lb} \in [0, 1)$ specifies the lower bound of the sampling probability, $gap \in [0, 1 - p_{lb}]$ controls the dynamic range, and $\kappa > 0$, $\mu \in (0, 1)$ are shape parameters governing the sharpness and center of the temporal transition. Intuitively, $p_+(t)$ biases online-source sampling toward later steps in the episode, whereas $p_-(t)$ emphasizes earlier steps. Larger values of $gap$ and $\kappa$ yield stronger and more abrupt temporal preferences, while smaller values result in more uniform sampling across steps. Based on this construction, we systematically analyze how model performance metrics respond to the effective online source ratio (OSR) under different temporal sampling regimes.

**Behaviour of** nlogi

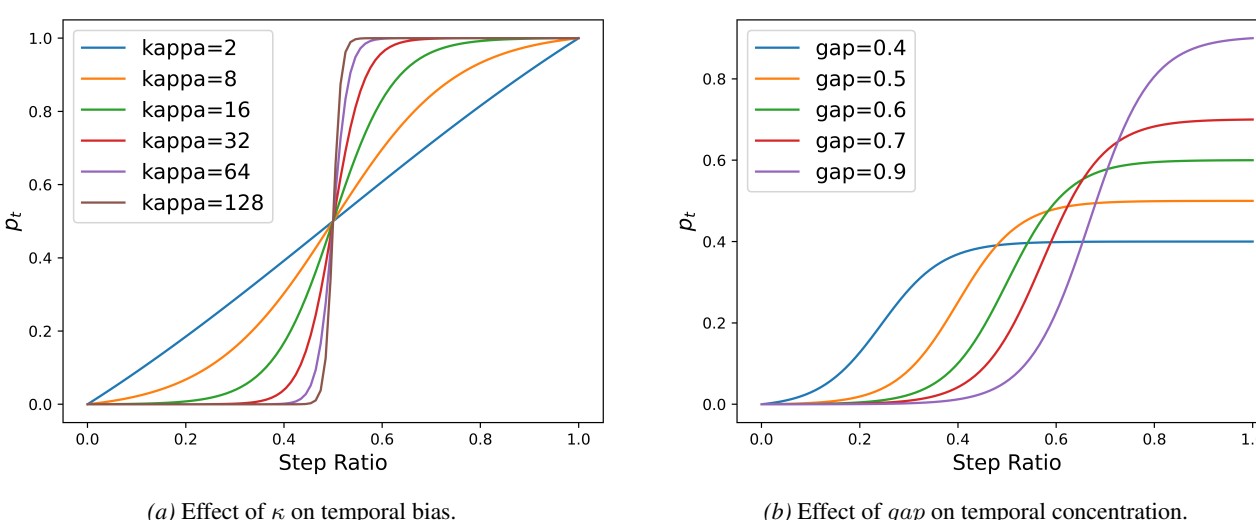

*(a)* Effect of $\kappa$ on temporal bias.

*(b)* Effect of $gap$ on temporal concentration.

*Figure 17.* Behavior of $\mathrm{nlogi}_+$ under different parameter settings. Left: Fixed $p_{lb} = 0$, $gap = 1$, and $\mathrm{E}(p) = 0.5$. Increasing $\kappa$ sharpens the temporal transition and induces stronger temporal bias, while smaller values lead to a more linear evolution. Right: Fixed $p_{lb} = 0$, $\kappa = 16$, and $\mathrm{E}(p) = 0.3$. Increasing $gap$ concentrates probability mass toward near-decision contexts, resulting in stronger temporal bias; smaller gaps yield a more temporally uniform distribution.

**Experimental Regime Construction.** Guided by Figure 17a, we fix $\kappa = 16$ to introduce a moderate level of nonlinearity in temporal bias. We discretize the interval $[0, 1]$ into four evenly spaced values. All ordered pairs $(p_{\mathrm{start}}, p_{\mathrm{end}})$ formed from these values yield $4 \times 4 = 16$ temporal sampling configurations.

The relative ordering between $p_{\mathrm{start}}$ and $p_{\mathrm{end}}$ determines the temporal regime: increasing, decreasing, or stationary on-track consistency. Specifically, this construction results in six increasing regimes, six decreasing regimes, and four stationary regimes. Within each regime, the absolute difference $|p_{\mathrm{start}} - p_{\mathrm{end}}|$ controls the strength of temporal bias.

Moreover, each $(p_{\mathrm{start}}, p_{\mathrm{end}})$ pair uniquely determines the admissible range of the expected online sampling probability $\mathbb{E}[p]$. For each configuration, we uniformly sample 50 values within this range. In total, this yields $16 \times 50 = 800$ experimental settings evaluated on the subset in Table 17, corresponding to 800 sampling points in the $(\mathrm{OSR}, \mathrm{Exact\ Match\ Ratio})$ plane.

### D.3. Sensitivity to On-Policy History for Qwen3-VL-8B-Thinking

The overall sensitivity of Qwen3-VL-8B-Thinking is consistent with that observed in GUI-Owl-7B.

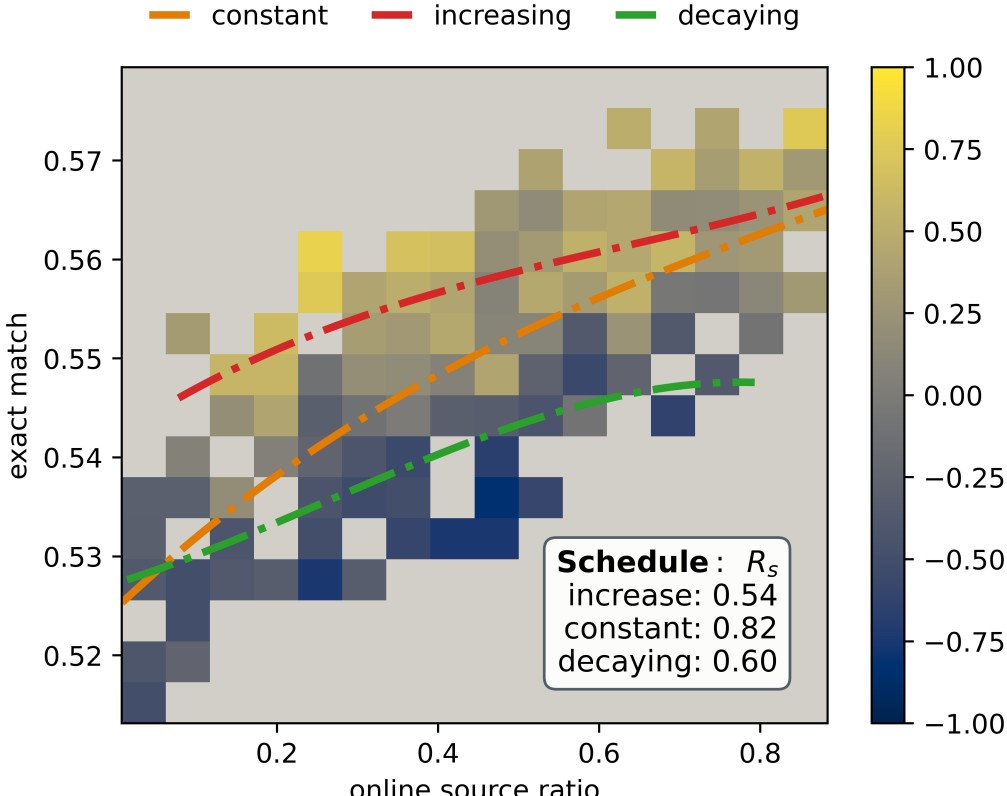

*Figure 18.* Relationship between exact match and online source ratio according to different on-policy mixing strategies for Qwen3-VL-8B-Thinking.

## D.4. Horizon-Length Degradation Analysis

We analyze step-level exact match as a function of both absolute step index and relative step ratio. The step-ratio stratification avoids conflating states that have the same absolute index but correspond to different task phases, such as early exploration in long episodes versus near-terminal states in short episodes. The evidence does not show a single universal cliff after a fixed number of steps. Instead, degradation is gradual and depends on both task phase and model family.

Across GUI-Owl-7B, UI-TARS-1.5-7B, and Qwen3-VL-8B-Thinking, early-to-mid task phases (step ratio $20\%$–$40\%$) already show degradation slightly before step 15, especially for longer tasks. In mid-phase tasks (step ratio $40\%$–$60\%$), the decline is milder and typically starts around step 15. GUI-specialized models remain comparatively stable through much of the mid-to-late regime, whereas Qwen3-VL-8B-Thinking shows a more monotonic decline from about step 5 onward. This pattern is consistent with a mixture of memory, search, and planning bottlenecks rather than a single horizon threshold.

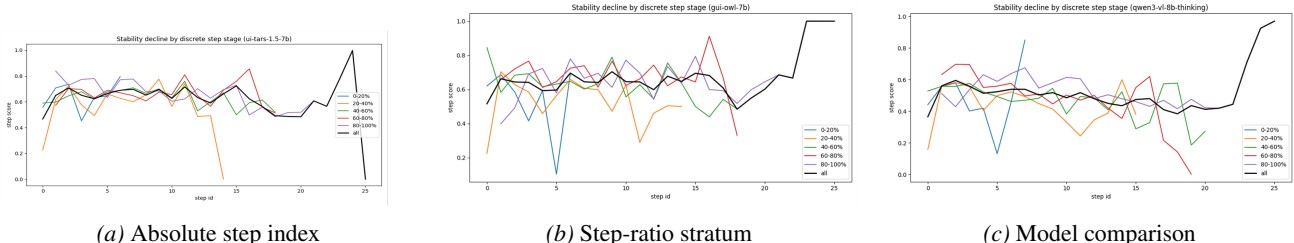

*(a)* Absolute step index   *(b)* Step-ratio stratum   *(c)* Model comparison

*Figure 19.* Horizon-length analysis using absolute step index and relative step ratio. The curves indicate phase- and model-dependent degradation rather than a universal sharp drop after a fixed step count.

### D.5. Case Study of Semi-Online Evaluation (SOEVAL)

Based on a qualitative inspection of representative step tasks, we identify two recurring patterns that characterize how SOEVAL alters decision stability under different task conditions.

### D.6. Enhanced Low-Baseline Stability Cases

This category primarily consists of step tasks that impose strict contextual constraints or require the integration of dispersed historical information. Typical examples include tasks restricting purchasing platforms, time-sensitive events, or the later stages of complex, multi-step objectives that may span multiple applications or sub-tasks.

Such tasks exhibit high information dispersion and often enforce cumulative value constraints, where the correctness of the current decision critically depends on accurately aggregating and interpreting prior context.

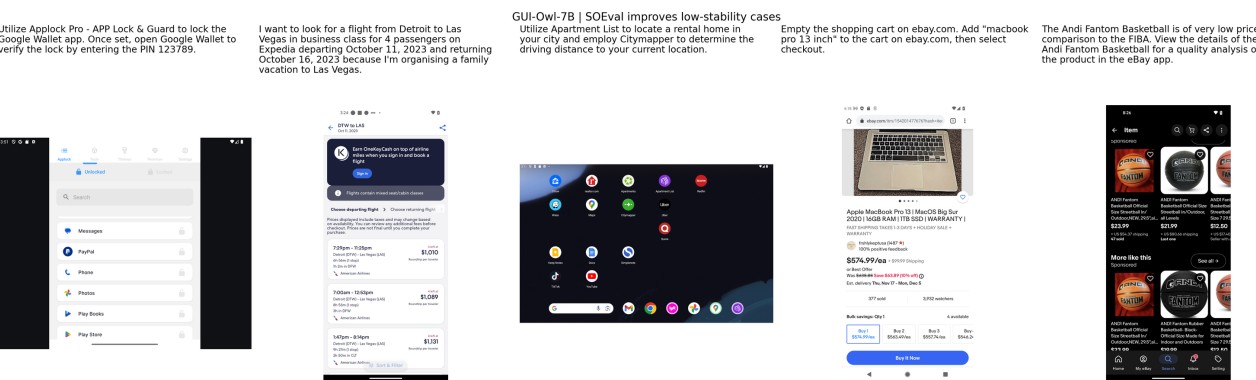

Enhanced Low-Baseline Stability Cases under SOEVAL.

### D.7. Suppressed High-Baseline Stability Cases

These tasks often correspond to the early stages of complex composite objectives, or to scenarios in which task completion can be reliably inferred from the final state alone, without requiring detailed historical context.

For such tasks, the offline evaluation setting already provides sufficient information to support confident decision-making. Introducing additional on-policy context via SOEVAL may therefore introduce superfluous or competing signals, slightly perturbing an otherwise stable decision distribution.

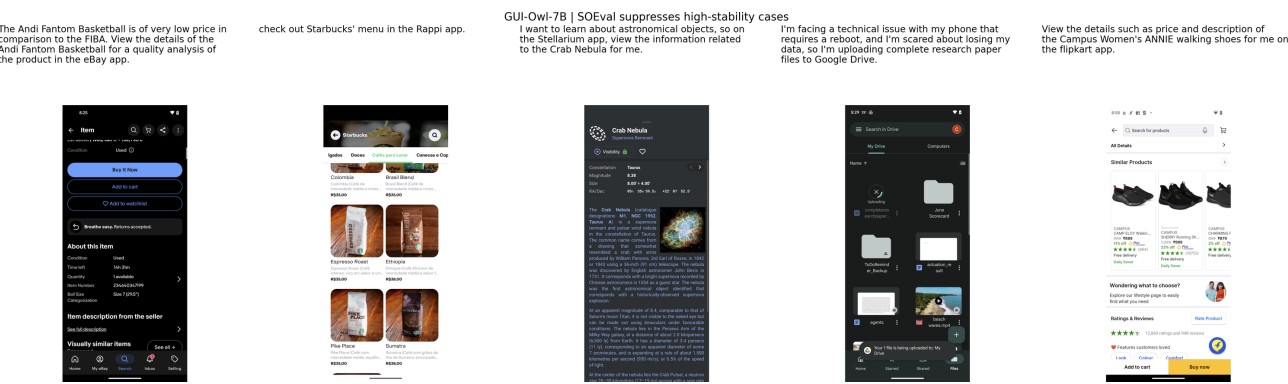

Suppressed High-Baseline Stability Cases under SOEVAL.

## E. Fixed-Thought Inference

For fixed-thought inference, we explicitly structure the message sequence by injecting a reasoning trace into the model context. Concretely, the message is organized as follows:

```
{
  ...,
  {
    'role': 'fixed_thought',
    'content': {'type': 'text', 'text': '{thought}'}
  }
}
```

To support this inference mode, we introduce only minimal modifications to the model's chat template, enabling the server-side prompt constructor to automatically incorporate the fixed thought into the prompt. The resulting prompt format used in our implementation is shown below:

### GUI-Owl-7B

```
{% set image_count = namespace(value=0) %}{% set video_count = namespace(value=0) %}{% for message in messages %}{%
    if loop.first and message['role'] != 'system' %}<|im_start|>system
You are a helpful assistant.<|im_end|>
{% endif %}{% if message['role'] != 'fixed_thought' %}<|im_start|>{{ message['role'] }}
{% if message['content'] is string %}{{ message['content'] }}<|im_end|>
{% else %}{% for content in message['content'] %}{% if content['type'] == 'image' or 'image' in content or '
    image_url' in content %}{% set image_count.value = image_count.value + 1 %}{% if add_vision_id %}Picture {{
    image_count.value }}: {% endif %}<|vision_start|><|image_pad|><|vision_end|>{% elif content['type'] == 'video'
    or 'video' in content %}{% set video_count.value = video_count.value + 1 %}{% if add_vision_id %}Video {{
    video_count.value }}: {% endif %}<|vision_start|><|video_pad|><|vision_end|>{% elif 'text' in content %}{{
    content['text'] }}{% endif %}{% endfor %}<|im_end|>
{% endif %}{% endif %}{% endfor -%}
<|im_start|>assistant
{%- if messages[-1].role == "fixed_thought" %}
<thinking>
{% if messages[-1].content is string %}{{messages[-1].content.strip('\n')}}{% else %}{% for content in messages[-1].
    content %}{% if content['type'] == 'text' %}{{ content['text'] }}{% endif %}{% endfor %}{% endif %}
</thinking>
{%- endif -%}
{{- "\n" -}}
```

### UI-Tars-1.5-7B

```
{% set image_count = namespace(value=0) %}{% set video_count = namespace(value=0) %}{% for message in messages %}{%
    if loop.first and message['role'] != 'system' %}<|im_start|>system
You are a helpful assistant.<|im_end|>
{% endif %}{% if message['role'] != 'fixed_thought' %}<|im_start|>{{ message['role'] }}
{% if message['content'] is string %}{{ message['content'] }}<|im_end|>
{% else %}{% for content in message['content'] %}{% if content['type'] == 'image' or 'image' in content or '
    image_url' in content %}{% set image_count.value = image_count.value + 1 %}{% if add_vision_id %}Picture {{
    image_count.value }}: {% endif %}<|vision_start|><|image_pad|><|vision_end|>{% elif content['type'] == 'video'
    or 'video' in content %}{% set video_count.value = video_count.value + 1 %}{% if add_vision_id %}Video {{
    video_count.value }}: {% endif %}<|vision_start|><|video_pad|><|vision_end|>{% elif 'text' in content %}{{
    content['text'] }}{% endif %}{% endfor %}<|im_end|>
{% endif %}{% endif %}{% endfor -%}
<|im_start|>assistant
{%- if messages[-1].role == "fixed_thought" %}
Thought: {% if messages[-1].content is string %}{{messages[-1].content.strip('\n')}}{% else %}{% for content in
    messages[-1].content %}{% if content['type'] == 'text' %}{{ content['text'] }}{% endif %}{% endfor %}{% endif
    %}
Action:
{%- endif -%}
```

### Qwen3-VL-8B-Thinking

```
{%- set image_count = namespace(value=0) %}
{%- set video_count = namespace(value=0) %}
{%- macro render_content(content, do_vision_count) %}
    {%- if content is string %}
        {{- content }}
    {%- else %}
        {%- for item in content %}
            {%- if 'image' in item or 'image_url' in item or item.type == 'image' %}
                {%- if do_vision_count %}
                    {%- set image_count.value = image_count.value + 1 %}
                {%- endif %}
                {%- if add_vision_id %}Picture {{ image_count.value }}: {% endif -%}
                <|vision_start|><|image_pad|><|vision_end|>
            {%- elif 'video' in item or item.type == 'video' %}
                {%- if do_vision_count %}
                    {%- set video_count.value = video_count.value + 1 %}
                {%- endif %}
                {%- if add_vision_id %}Video {{ video_count.value }}: {% endif -%}
                <|vision_start|><|video_pad|><|vision_end|>
```

```
                {%- elif 'text' in item %}
                    {{- item.text }}
                {%- endif %}
            {%- endfor %}
        {%- endif %}
{%- endmacro %}
{%- if tools %}
    {{- '<|im_start|>system\n' }}
    {%- if messages[0].role == 'system' %}
        {{- render_content(messages[0].content, false) + '\n\n' }}
    {%- endif %}
    {{- "# Tools\n\nYou may call one or more functions to assist with the user query.\n\nYou are provided with
     function signatures within <tools></tools> XML tags:\n<tools>" }}
    {%- for tool in tools %}
        {{- "\n" }}
        {{- tool | tojson }}
    {%- endfor %}
    {{- "\n</tools>\n\nFor each function call, return a json object with function name and arguments within <
     tool_call></tool_call> XML tags:\n<tool_call>\n{\"name\": <function-name>, \"arguments\": <args-json-object>}\n
     </tool_call><|im_end|>\n" }}
{%- else %}
    {%- if messages[0].role == 'system' %}
        {{- '<|im_start|>system\n' + render_content(messages[0].content, false) + '<|im_end|>\n' }}
    {%- endif %}
{%- endif %}
{%- set ns = namespace(multi_step_tool=true, last_query_index=messages|length – 1) %}
{%- for message in messages[::-1] %}
    {%- set index = (messages|length – 1) – loop.index0 %}
    {%- if ns.multi_step_tool and message.role == "user" %}
        {%- set content = render_content(message.content, false) %}
        {%- if not(content.startswith('<tool_response>') and content.endswith('</tool_response>')) %}
            {%- set ns.multi_step_tool = false %}
            {%- set ns.last_query_index = index %}
        {%- endif %}
    {%- endif %}
{%- endfor %}
{%- for message in messages %}
    {%- set content = render_content(message.content, True) %}
    {%- if (message.role == "user") or (message.role == "system" and not loop.first) %}
        {{- '<|im_start|>' + message.role + '\n' + content + '<|im_end|>' + '\n' }}
    {%- elif message.role == "assistant" %}
        {%- set reasoning_content = '' %}
        {%- if message.reasoning_content is string %}
            {%- set reasoning_content = message.reasoning_content %}
        {%- else %}
            {%- if '</think>' in content %}
                {%- set reasoning_content = content.split('</think>')[0].rstrip('\n').split('<think>')[-1].lstrip('\
n') %}
                {%- set content = content.split('</think>')[-1].lstrip('\n') %}
            {%- endif %}
        {%- endif %}
        {%- if loop.index0 > ns.last_query_index %}
            {%- if loop.last or (not loop.last and reasoning_content) %}
                {{- '<|im_start|>' + message.role + '\n<think>\n' + reasoning_content.strip('\n') + '\n</think>\n\n'
     + content.lstrip('\n') }}
            {%- else %}
                {{- '<|im_start|>' + message.role + '\n' + content }}
            {%- endif %}
        {%- else %}
            {{- '<|im_start|>' + message.role + '\n' + content }}
        {%- endif %}
        {%- if message.tool_calls %}
            {%- for tool_call in message.tool_calls %}
                {%- if (loop.first and content) or (not loop.first) %}
                    {{- '\n' }}
                {%- endif %}
                {%- if tool_call.function %}
                    {%- set tool_call = tool_call.function %}
                {%- endif %}
                {{- '<tool_call>\n{"name": "' }}
                {{- tool_call.name }}
                {{- '", "arguments": ' }}
                {%- if tool_call.arguments is string %}
                    {{- tool_call.arguments }}
                {%- else %}
                    {{- tool_call.arguments | tojson }}
                {%- endif %}
                {{- '}\n</tool_call>' }}
            {%- endfor %}
        {%- endif %}
        {{- '<|im_end|>\n' }}
```

```
    {%- elif message.role == "tool" %}
        {%- if loop.first or (messages[loop.index0 - 1].role != "tool") %}
            {{- '<|im_start|>user' }}
        {%- endif %}
        {{- '\n<tool_response>\n' }}
        {{- content }}
        {{- '\n</tool_response>' }}
        {%- if loop.last or (messages[loop.index0 + 1].role != "tool") %}
            {{- '<|im_end|>\n' }}
        {%- endif %}
    {%- endif %}
{%- endfor %}
<|im_start|>assistant
<think>
{%- if messages[-1].role == "fixed_thought" %}
{% if messages[-1].content is string %}{{messages[-1].content.strip('\n')}}{% else %}{% for content in messages[-1].
    content %}{% if content['type'] == 'text' %}{{ content['text'] }}{% endif %}{% endfor %}{% endif %}
</think>
{% endif -%}
{{- "\n" -}}
```

## F. Reasoning-Execution Consistency Analysis Details

Tables 21 and 22 report the balanced confusion matrix and Wilson confidence intervals used to characterize the residual noise of the two-stage reasoning-execution consistency detector.

| Human label | Pred. consistent | Pred. inconsistent | Total |
|---|---|---|---|
| Consistent | 273 | 51 | 324 |
| Inconsistent | 15 | 309 | 324 |
| Total | 288 | 360 | 648 |

*Table 21.* Balanced confusion matrix for the two-stage reasoning-execution consistency detector. The human label *Consistent* corresponds to reasoning–execution consistency.

| Metric | Estimate | Wilson 95% CI |
|---|---|---|
| Accuracy | 0.898 | [0.873, 0.919] |
| TPR | 0.843 | [0.799, 0.878] |
| TNR | 0.954 | [0.925, 0.972] |

*Table 22.* Wilson confidence intervals for two-stage R-E consistency detection.

The examples below align the qualitative R-E consistency cases with the failure taxonomy in Table 7. Figure 20 illustrates an action-target mismatch: the execution uses the intended interaction type but selects the wrong target, corresponding to a grounding-oriented failure. Figure 21 illustrates an action-type mismatch: the reasoning intent is not reflected in the executed interaction type, corresponding to a planning/action-selection failure.

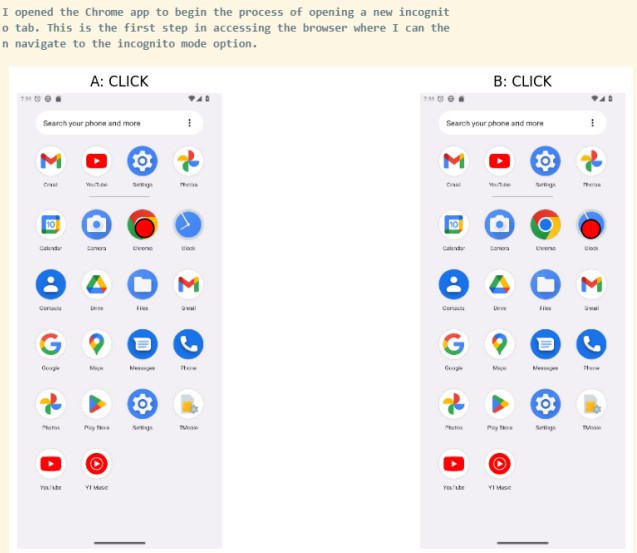

*Figure 20.* Execution target mismatch, corresponding to the action-target category in the R-E consistency failure taxonomy.

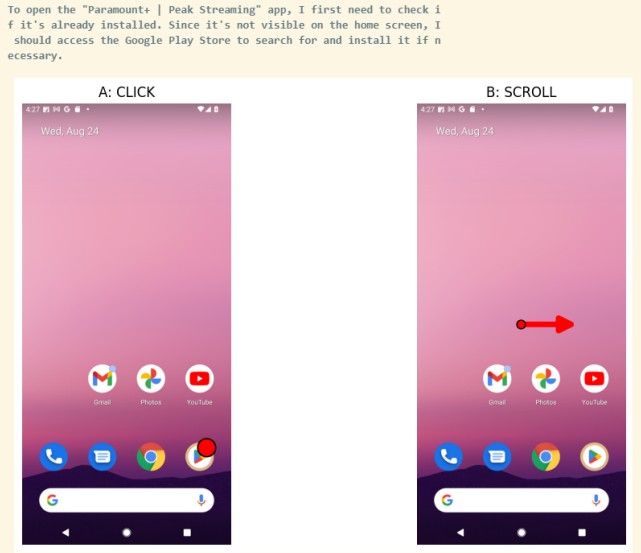

*Figure 21.* Execution type mismatch, corresponding to the action-type category in the R-E consistency failure taxonomy.

