# OpenReview forum: "Scaling, Benchmarking, and Reasoning of Vision-Language Agents for Mobile GUI Navigation"
_ICML.cc/2026/Conference — ICML 2026 regular_

### Official Review · Reviewer_FdMh · 2026-03-06

**Soundness:** 3
**Presentation:** 4
**Significance:** 3
**Originality:** 2
**Overall Recommendation:** 4
**Confidence:** 5

**Summary:**

This paper studies vision-language agents for mobile GUI navigation, focusing on scaling behavior, benchmarking, and reasoning capabilities. The authors investigate how multimodal agents interact with mobile interfaces through perception ad screen understanding, language-conditioned goal interpretation, and action generation. They analyze how performance scales with model size, training data, and interaction trajectories, and introduce or extend benchmarks that better reflect realistic mobile task completion scenarios. The work emphasizes reasoning over GUI elements like icons, layouts, dynamic states rather than pure visual recognition and evaluates agents on multistep goal driven tasks requiring state tracking, planning and error recovery. Through empirical experiments the paper identifies performance bottlenecks in current vision-language agents operating in mobile environments.

**Compliance With Llm Reviewing Policy:**

Affirmed.

**Key Questions For Authors:**

Few questions to address:
1. How well do agents generalize to unseen apps, UI layouts, or OS versions? Are results primarily in-distribution?
2. How does performance degrade as task horizon length increases? Is there a sharp drop after a certain number of steps? -> Basically, I want to understand the memory and planning bottlenecks.
3. Can you quantitatively separate failures caused by visual grounding errors from reasoning/planning failures?

One prior work have addressed the OS generalization: AppAgent: Multimodal Agents as Smartphone Users, OS-ATLAS: A Foundation Action Model for Generalist GUI Agents
One prior works have addressed the planning v/s action failures: OmniACT: A Dataset and Benchmark for Enabling Multimodal Generalist Autonomous Agents for Desktop and Web

**Limitations:**

The paper discusses experimental constraints but could more explicitly acknowledge limitations around simulator fidelity, annotation assumptions, and action space simplifications. GUI benchmarks often rely on static screenshots or semi-deterministic simulators, which may not capture real-world latency or dynamic UI updates or unexpected pop-ups. Additionally, scaling results may be sensitive to dataset composition and task distribution; more clarity on these dependencies would improve transparency. Papers like WebArena: A Realistic Web Environment for Building Autonomous Agents do that for web. From a societal perspective, mobile GUI automation introduces both opportunities and risks. While such agents could improve accessibility and productivity, they may also enable automated misuse (for example spam, fraudulent interactions or bypassing user safeguards). The authors should discuss responsible deployment, safeguards, and evaluation metrics aligned with safety and robustness, particularly since mobile environments often involve sensitive personal data.

**Strengths And Weaknesses:**

**Soundness**

The experimental setup appears methodologically solid with structured benchmarking across multiple tasks and scaling regimes. The author's empirical claims about scaling behavior and reasoning limitations are generally supported by experiments. The analysis of stepwise success rates, trajectory completion, and reasoning breakdowns strengthens the technical credibility. However, the authors could improve by providing deeper ablations isolating perception v/s planning errors and clarifying whether improvements come from better visual grounding or stronger language priors.


**Presentation**
The paper is generally well structured and easy to follow. The separation between scaling analysis, benchmarking framework, and reasoning evaluation is clear. Figures illustrating GUI trajectories and failure cases are particularly helpful. The overall writing and presentation is really great. I like the comprehensive figures presented by the authors. I would recommend the authors to cite some more relevant works to make the literature review sound.

**Significance**
This problem is highly relevant. Mobile GUI automation is a practical and commercially important setting for multimodal agents. Unlike static VQA type of tasks GUI navigation requires perception, grounding, memory, and multistep reasoning under changing states. The scaling analysis is particularly valuable. Understanding how performance grows with model capacity and data is critical for guiding future agent development. If the benchmark becomes widely adopted, this work could meaningfully influence research on embodied and interactive vision-language agents.

**Originality**

While GUI navigation itself is not entirely new, the author's contribution lies in systematically studying scaling laws and reasoning behaviors in this domain. The explicit focus on reasoning diagnostics and performance scaling across model regimes provides new empirical insight. The originality is not in a radically new model architecture, but in framing and evaluating GUI agents through scaling and reasoning lenses. This analytical perspective adds meaningful value beyond incremental benchmark extensions.

---

> ### Author Rebuttal · Authors · 2026-03-31
>
> # Response to Reviewer FdMh
>
> We thank the reviewer for the careful reading and the concrete questions.
>
> ## For Concerns from Soundness and Question 3
> We thank the reviewer for this suggestion.
>
> We manually annotated a subset of thought–execution inconsistent cases (n=108) into three categories:
>
> | Error Type | Count | Ratio | Description |
> | --- | --- | --- | --- |
> | Action-type mismatch | 66 | 61.1% | Correct intent but wrong interaction type (planning error) |
> | Action-target mismatch | 40 | 37.0% | Correct interaction type but wrong target (grounding error) |
> | Invalid action | 2 | 1.9% | No valid interaction |
>
> The results suggest that planning-level errors (action-type mismatch) are the dominant source of failure, outweighing pure visual grounding errors by a ratio of roughly 5:3. We will include this breakdown and qualitative examples in the revision.
>
> Our reasoning–execution consistency (TEC) analysis provides a partial answer: among thought–execution inconsistent cases, decision/action-selection mismatches (planning errors) are more common than pure grounding mismatches. However, we agree this is failure-mode attribution rather than a fully controlled isolation of visual grounding vs. language priors. In the revision, we will add a finer-grained error breakdown by categorizing incorrect predictions into: (1) grounding errors (correct action type but wrong spatial target) and (2) planning errors (wrong action type or target element). This will more directly address whether improvements stem from perception or reasoning capabilities.
>
> ## For Concerns from Presentation (Literature Review)
> We will cite the suggested works (AppAgent, OS-ATLAS, and OmniACT) in the revised literature review and discuss their relevance to OS generalization and planning vs. action failure analysis, respectively.
>
> ## For Question 1
> Our evaluation explicitly addresses generalization through four carefully designed test splits (Figure 1): (1) IDD (in-distribution), with tasks from the same apps as training; (2) Unseen App, with tasks from apps not present in training; (3) Unseen Device, with tasks collected on tablet devices (training uses only mobile phones); and (4) Unseen App & Device, combining both. As shown in Figure 1, results are reported across all four splits. Notably, RL-trained models show substantially stronger generalization than SFT-trained models in out-of-distribution settings, demonstrating that our analysis goes well beyond in-distribution evaluation.
>
> ## For Question 2
> We analyzed step-level exact match as a function of both absolute step index and relative step ratio (step_id / episode_length) to avoid conflating different task phases.
> Key findings across GUI-Owl-7B, UI-TARS-1.5-7B, and Qwen3-VL-8B-Thinking:
> - No single sharp drop-off. Degradation is gradual and phase-dependent rather than a cliff after a fixed number of steps.
> - Degradation begins earlier for longer tasks. In early-phase steps (step ratio 20–40%) of long episodes (>35 steps), performance degrades noticeably before step 15, likely due to broader search scope and heavier memory demands.
> - Model-family differences. GUI-specialized models (GUI-Owl-7B, UI-TARS-1.5-7B) maintain relatively stable performance (~63% exact match) through mid-to-late phases before step 15. In contrast, Qwen3-VL-8B-Thinking shows more monotonic degradation from step 5 onward, suggesting greater sensitivity to horizon length.
>
> These patterns suggest a mixture of memory and planning bottlenecks rather than a single capacity threshold. We will include this analysis with supporting figures in the revision.
>
>
> ## For Concerns on Limitations of Proposed Evaluation Scheme and Setup
> We agree that SOEval is not true online evaluation and does not fully capture real world dynamics.
>
> For detailed discussion and quantitative evidence on its relation to online metrics, please see our **Response to Reviewer 1nSs**, _For Weakness 1, 3 and Question 1_.
>
>
> ## For Ethical Concerns on Potential Misuse, Privacy, and Responsible Deployment
> >From a societal perspective, mobile GUI automation introduces both opportunities and risks. While such agents could improve accessibility and productivity, they may also enable automated misuse (for example spam, fraudulent interactions or bypassing user safeguards)...
>
> We fully agree that mobile GUI automation carries both societal benefits and potential risks. During the construction of HyperTrack, we applied privacy safeguards, including manual review and discarding data containing personally identifiable information. We also avoided collecting data from apps primarily handling highly sensitive information. In the revision, we will add an Impact Statement section discussing responsible deployment safeguards, potential misuse risks, and safety-aligned evaluation directions for future work.

---

> > ### Author Rebuttal · Reviewer_FdMh · 2026-04-06
> >
> > Looks good from my standpoint. Thanks for resolving the questions

---

> > > ### Author Response · Authors · 2026-04-06
> > >
> > > We are glad to see that we have addressed your concerns, and we sincerely appreciate your constructive suggestions that helped improve our work.

---

### Official Review · Reviewer_1nSs · 2026-03-11

**Soundness:** 3
**Presentation:** 3
**Significance:** 2
**Originality:** 2
**Overall Recommendation:** 5
**Confidence:** 4

**Summary:**

The paper presents HyperTrack, a large-scale Chinese mobile GUI navigation dataset with 16k tasks across 674 apps, and GUIEvalKit, a unified benchmarking toolkit supporting both offline and semi-online evaluation for vision-language GUI agents. Using HyperTrack, the authors study data scaling effects under supervised vs. reinforcement-based finetuning, finding consistent gains for RL especially out-of-domain, and they provide a broad benchmark of state-of-the-art VLM agents alongside a decision-level analysis of how interaction history and explicit reasoning affect task performance. The work introduces decision diversity and stability metrics, demonstrates that “thinking” often reduces PASS@1 but can outperform with larger rollout budgets, and shows SOEval improves stability for previously unstable decisions by conditioning on on-policy history.

**Compliance With Llm Reviewing Policy:**

Affirmed.

**Final Justification:**

The rebuttal addressed my concerns.

**Key Questions For Authors:**

- How does SOEval correlate with actual online success across a representative subset of tasks and models? Can you report R^2 compared to offline evaluation to substantiate its claimed fidelity benefits?
- How robust are the decision diversity/stability conclusions to the clustering method and hyperparameters? Please include sensitivity analyses (e.g., different distance metrics, neighborhood thresholds) and measures of cluster stability.
- Can you add episode-level success/progress metrics with confidence intervals across multiple seeds to complement the step-level metrics reported?
- Can you explain the absence of correlation to true online evaluation, limited ablations on the RL objective, and incomplete positioning relative to closely related recent work (UItron, UI-S1, MobiBench)?

**Limitations:**

yes

**Strengths And Weaknesses:**

Strengths
- Introduces a semi-online evaluation protocol (SOEval) that systematically integrates on-policy reasoning/history when the model stays on-track, bridging a gap between purely offline and online evaluation.
- Proposes decision-level analysis metrics (decision diversity and decision stability) based on clustering execution samples to disentangle robustness vs. exploration, providing actionable insights into reasoning-induced variance.
- Presents a large, Chinese-centric mobile GUI dataset with hierarchical UI docs, precise bounding boxes, and screen descriptions—filling a notable gap in language and app coverage.
- Broad benchmarking across multiple established datasets and a wide range of VLMs and GUI-specialized agents, including reasoning vs. instruct-only modes.

Weaknesses
- The SOEval procedure, while useful, still relies on the “on-track” assumption and static reference trajectories; its correlation to real online success is not quantified against true online rollouts.
- The decision clustering that induces “semantic decisions” can be sensitive to clustering choices and distance metrics; robustness analyses are not reported.
- No direct comparison or correlation study between SOEval and online evaluations to validate that SOEval better predicts online performance than standard offline evaluation.
- The curated reasoning-execution consistency dataset and the two-stage voting reach ~0.90 accuracy but introduce noise; confidence intervals and error analyses are limited.
- Scaling study is focused on one backbone (UI-TARS-1.5-7B); it would be stronger to replicate with another model family to establish generality.
- Episode-level outcomes (progress/success) are referenced but the main tables emphasize step-level metrics; more thorough episode-level reporting and variance across seeds would help.
- Overlaps with UItron’s GRPO-based RL training and large-scale Chinese app data are not discussed; a direct comparison or at least positioning would clarify novelty.

---

> ### Author Rebuttal · Authors · 2026-03-31
>
> # Response to Reviewer 1nSs
>
> We thank the reviewer for the careful reading and the concrete questions.
>
> ## For Weakness 1, 3 and Question 1
> We agree that SOEval is not true online evaluation and does not fully capture trajectory-level error accumulation or long-horizon drift; we will state this limitation explicitly in the revision.
>
> Our goal is narrower: SOEval is a step-level, context-aligned protocol that reduces the mismatch between offline evaluation (conditioned on reference history) and online inference (conditioned on model-generated history). It uses on-policy context when the model stays on-track and falls back to the reference trajectory otherwise, trading full online fidelity for stable step-level analysis.
>
> Empirically, we report the correlation between evaluation metrics across six models:
>
> | Metric | Qwen2.5-VL-3B-Instruct | Qwen3-VL-4B-Thinking | Qwen3-VL-8B-Instruct | GUI-Owl-7B | GUI-Owl-32B | UI-Venus-Navi-72B | $R^2$ | $Spearman\ \rho$ |
> | ----------- | ----------- | ----------- | ----------- | ----------- | ----------- | ----------- | ----------- | ----------- |
> | SOEval (Exact Match) | 55.93 | 57.66 | 62.84 | 67.37 | 71.66 | 76.16 | 0.6241 | 0.7714 |
> | Offline (Exact Match) | 56.68 | 54.52 | 60.39 | 65.49 | 70.95 | 74.09 | 0.4821 | 0.6571 |
> | SOEval (Progress) | 8.19 | 8.40 | 9.00 | 12.01 | 14.17 | 15.84 | 0.5377 | 0.7714 |
> | AW (OnlineSuccess) | 13.0 | 52.0 | 47.6 | 66.4 | 67.0 | 65.9 | - | - |
>
> SOEval exact match achieves $R^2=0.6242$ (Spearman $\rho=0.7714$) with online success, compared to $R^2=0.4821$ (Spearman $\rho=0.6571$) for standard offline evaluation. This confirms that SOEval better approximates online performance at the step level, consistent with its design as a step-level local approximation rather than a full online rollout simulation.
>
> ## For Weakness 2 and Question 2
> We conducted sensitivity analyses varying DBSCAN $\varepsilon \in \{5, 15, 30, 50, 70, 90, 100, 140\}$ and comparing $L_1$ vs. $L_2$ distance metrics on all three models.
>
> The clustering structure is stable across $\varepsilon$: cross-$\varepsilon$ supporting-size correlation is high ($R^2=0.95$ for $\varepsilon=25$ vs. $70$; $R^2=0.99$ for $\varepsilon=70$ vs. $100$), and the normalized Wasserstein distance between $\varepsilon$-induced decision distributions remains low and nearly constant ($\sim 0.163$) over $\varepsilon=5 \rightarrow 140$, indicating local refinement rather than structural reorganization. After scale adjustment $\hat \varepsilon_2 = \frac{\sqrt 2 + 1}{2} \varepsilon_1$, $L_1$ and $L_2$ yield nearly identical decision distributions (Wasserstein distance < $5e-4$). Under both $\varepsilon=30$ and $\varepsilon=140$, the diversity–stability trade-off remains qualitatively unchanged, with only minor count-level variation (typically < $\pm 5%$) relative to the baseline $\varepsilon=70$.
>
> We will include the full sensitivity tables and analysis in the revision appendix.
>
> ## For Weakness 4
> We agree and will add confidence intervals and a brief error analysis in the revision. On the curated benchmark (300 consistent + 300 inconsistent), the two-stage voting detector achieves 0.898 accuracy with 95% CI [0.872, 0.920], TPR 0.843 [0.798, 0.880], and TNR 0.954 [0.923, 0.972]. Despite residual noise, the downstream association remains strong: execution success is 73.7% for consistency-positive samples versus 18.3% for consistency-negative ones, confirming that the main conclusion is robust.
>
> ## For Weakness 5
> We have added a complementary scaling analysis on a second model family, which supports the same qualitative conclusion that RL scales more favorably than SFT, especially in out-of-domain settings.
>
> For detailed evidence, please see our **Response to Reviewer eVUL**, _For Concern 2_, where we broaden the evidence along the two axes raised, namely model family and reward definition, while showing that the main conclusions continue to hold.
>
> ## For Weakness 6 and Question 3
> We agree that episode-level reporting should be made more explicit. On the decision-level evaluation subset, episode-level progress over 8 random seeds shows low variation across three representative models (95% CI half-width: 0.0062 for Qwen3-VL-8B-Thinking, 0.0030 for GUI-Owl-7B, and 0.0058 for UI-Tars-1.5). We will include these multi-seed subset results in the revision to complement the step-level metrics; full multi-seed evaluation over the entire benchmark suite is substantially more expensive.
>
> ## For Weakness 7 and Question 4
> We will revise the related-work discussion accordingly. In brief, UItron focuses on training an RL-based Chinese GUI agent, while our paper centers on (i) a systematic SFT-vs.-RL scaling study, (ii) a unified benchmarking toolkit spanning multiple benchmarks and model families, and (iii) decision-level analysis of reasoning and context construction. We will make this distinction explicit and discuss UI-S1 and MobiBench as complementary evaluation settings.

---

> > ### Author Rebuttal · Reviewer_1nSs · 2026-04-04
> >
> > Thanks for the detailed responses. My questions are fully reolved. I will adjust the score.

---

> > > ### Author Response · Authors · 2026-04-04
> > >
> > > We are glad to see that we have addressed your concerns, and we sincerely appreciate your support in raising the score, as well as your constructive suggestions that helped improve our work.

---

### Official Review · Reviewer_Rdf3 · 2026-03-12

**Soundness:** 2
**Presentation:** 3
**Significance:** 2
**Originality:** 2
**Overall Recommendation:** 3
**Confidence:** 5

**Summary:**

This submission presents a systematic study of data scaling, benchmarking, and reasoning for VLM-based agents in mobile GUI navigation. They introduce a large-scale dataset  named HyperTrack, with over 16000 real-world tasks across more than 650 Chinese mobile applications. They also develop GUIEvalKit, an open-source toolkit for unified benchmarking of VLMs on offline GUI navigation tasks. Moreover, they explore supervised finetuning and reinforcement-based finetuning, which highlights the synergy between data scaling and reinforcement learning.

**Compliance With Llm Reviewing Policy:**

Affirmed.

**Final Justification:**

I believe the author's improvements to the presentation would enhance the paper's quality, therefore I opted for a higher score. However, after carefully reading the author's response, I still lean towards a weak rejection, as I don't believe this work is significant enough to inspire future work. I don't think this paper is worthy of publication at ICML.

**Key Questions For Authors:**

please see  weaknesses

**Limitations:**

yes

**Strengths And Weaknesses:**

Strengths:

--This work contributes a systematic study of data scaling, benchmarking, and reasoning for GUI navigation

--This work conducts a large-scale Chinese dataset of 16000 real-world tasks and an open-source toolkit named GUIEvalKit

Weaknesses:

--Soundness: The technically sound of this paper is very poor. This paper does not contribute any new methods. The evaluation tools and conclusions it offers are unimportant; there is insufficient evidence to support their necessity for the field.

--Presentation: This paper is poorly written. The paper's technical contributions are vague, the description of related work is insufficient, and the contents of Tables 1 and 2 are difficult to understand.

--Significance: The problem addressed in this paper is relatively unimportant in the field of GUI agents because it only studies navigation in a limited offline environment, lacking crucial research in online environments. More importantly, the paper's comparison of reinforcement learning and SFT is already well-known.

--Originality: This paper lacks novelty and originality. The authors did not contribute any new data construction methods or training algorithms, nor even new evaluation methods. The core of the paper is merely to construct a relatively unimportant dataset and demonstrate some well-known conclusions by reproducing open-source work.

---

> ### Author Rebuttal · Authors · 2026-03-31
>
> # Response to Reviewer Rdf3
>
> We thank the reviewer for reading the paper and for highlighting concerns about soundness, presentation, significance, and originality.
>
> ## For Concerns on Soundness
> We respectfully disagree with this assessment. The paper's contribution is not a new policy architecture, but a systematic empirical and evaluation study.
>
> Concretely, the paper contributes both HyperTrack (a large-scale Chinese GUI dataset with features absent from existing datasets) and GUIEvalKit (a unified toolkit standardizing GUI evaluation). The paper also contributes new methodology on the evaluation side: (1) SOEval, a semi-online protocol that recovers on-policy context to reduce evaluation mismatch, with empirically stronger correlation to online performance than standard offline evaluation; (2) decision diversity and decision stability, new metrics that reveal a stability–diversity trade-off, reasoning broadens viable decisions but reduces PASS@1, a finding not previously characterized in GUI agents; and (3) reasoning–execution consistency analysis, showing that thought–action mismatches strongly correlate with execution failure.
>
> Regarding the RL vs. SFT comparison: existing GUI agent RL work trains RL at a single fixed data scale and demonstrates that RL with small data can match or surpass SFT with larger data. However, no prior work has systematically studied how RL scales with training data size in this domain. Our work fills this gap by conducting the first RL scaling study, revealing the log-linear scaling trend and RL's growing advantage over SFT.
>
> ## For Concerns on Presentation
> We agree the contribution type should be stated more explicitly. The paper contributes (1) a large-scale Chinese mobile GUI dataset, (2) a unified benchmarking toolkit, (3) a controlled scaling study of SFT vs. RL, and (4) an analysis framework for reasoning and evaluation-context effects. We will revise the introduction and conclusion to make this distinction sharper.
>
> We will expand the literature review, especially around GUI-agent evaluation, offline vs. online assessment, and recent RL-based GUI agent work (UItron, UI-S1, MobiBench, AppAgent, OS-ATLAS, etc.).
>
> We will improve readability for both tables. For Table 1, we will expand the caption to explain each column (e.g., "Hier. docs" = hierarchical UI document annotations) so the cross-dataset comparison is immediately clear. For Table 2, we will add an explicit note explaining the "X / Y" format (step type match / step exact match) and introduce visual grouping to separate open-source VLMs from GUI-specialized agents.
>
> ## For Concerns on Significance
> We respectfully disagree with this assessment.
>
> The paper does not claim offline evaluation is sufficient — in fact, this limitation is precisely our motivation for proposing SOEval, which bridges offline and online evaluation by restoring on-policy context. Full online evaluation is costly and hard to standardize across models and benchmarks; SOEval addresses a concrete and practical part of this gap. We also provide empirical evidence that SOEval better correlates with online performance than standard offline evaluation.
>
> Existing GUI agent RL work usually trains RL at a single fixed data scale. Our work provides the first RL scaling study in this domain, revealing that RL's advantage grows substantially, especially in out-of-domain settings, a finding that cannot be obtained from single-scale comparisons.
>
> ## For Concerns on Originality
> We believe this applies an overly narrow definition of originality. The paper's contributions are original on the evaluation and analysis side:
> SOEval is a new evaluation protocol — not a restatement of offline evaluation. It explicitly incorporates on-policy decision artifacts when the model stays on-track, producing measurably better correlation with online performance.
>
> Decision diversity and decision stability are new analytical metrics that reveal a previously uncharacterized stability–diversity trade-off: reasoning broadens viable decisions but reduces PASS@1, while improving PASS@n under more tolerant evaluation.
>
> To the best of our knowledge, the RL scaling study is the first in the GUI agent domain to systematically vary training data size for RL, as no prior work has done so.
>
> As Reviewer FdMh also noted: "the originality is not in a radically new model architecture, but in framing and evaluating GUI agents through scaling and reasoning lenses. This analytical perspective adds meaningful value beyond incremental benchmark extensions."

---

> > ### Author Rebuttal · Reviewer_Rdf3 · 2026-04-03
> >
> > Thank you for the reply and clarifications. I now have a better understanding of the motivation behind this work.
> > I believe the author's improvements to the presentation will enhance the quality of the paper.
> > Therefore, I have raised my score.

---

> > > ### Author Response · Authors · 2026-04-04
> > >
> > > We are glad to see that we have addressed your concerns, and we sincerely appreciate your recognition of the clarification and presentation improvements, as well as your support in raising the score.
> > >
> > > At the same time, we would greatly appreciate any further specific concerns you may still have. We would be more than happy to clarify them in order to make additional improvements on our work and  the rating where possible.

---

### Official Review · Reviewer_eVUL · 2026-03-12

**Soundness:** 3
**Presentation:** 2
**Significance:** 2
**Originality:** 3
**Overall Recommendation:** 4
**Confidence:** 4

**Summary:**

This paper studies mobile GUI navigation with VLM agents from three angles: dataset scale, benchmarking, and reasoning behavior. It introduces HyperTrack, a Chinese mobile GUI dataset with 16,080 tasks from 674 Android apps, and GUIEvalKit, a unified evaluation toolkit spanning five benchmarks and more than 30 models. The paper also presents a scaling study comparing SFT and RL on UI-TARS-1.5-7B, proposes a semi-online evaluation protocol (SOEval), and analyzes a stability–diversity trade-off for reasoning-enabled agents.

**Compliance With Llm Reviewing Policy:**

Affirmed.

**Final Justification:**

I maintain my score.

**Key Questions For Authors:**

see above

**Limitations:**

No. see above

**Strengths And Weaknesses:**

Concerns:
1. The dataset release and data quality story feels incomplete in the visible main text. The paper describes the annotation format, but I did not see strong evidence on annotation quality control, inter-annotator agreement, privacy filtering, or app redistribution/reproducibility constraints. For a dataset paper, those details matter a lot.
2.  The RL scaling conclusion is plausible but somewhat narrow experimentally. The scaling study fine-tunes a single base model, UI-TARS-1.5-7B, under one reward design and one RL recipe. It is not clear how broadly the result extends across model families or reward definitions.
3. The author said it is a Chinese mobile GUI dataset, however, none of the sample/screenshot in appendix is Chinese.
4. SOEval is useful, but still not true online evaluation. It improves the evaluation context by conditioning on model-generated histories, but it does not fully capture real rollout errors and trajectory drift. The paper should clarify this limitation more explicitly.

---

> ### Author Rebuttal · Authors · 2026-03-31
>
> # Response to Reviewer eVUL
>
> We thank the reviewer for the careful reading and the concrete questions.
>
> ## For Concern 1
> HyperTrack was created with multiple trained annotators using a custom annotation tool that enforces the structured action space and supports interactive bounding-box annotation, reducing annotation errors at the source. Completed episodes were then cross-reviewed by a separate group for task coherence, action correctness, and box accuracy, with failed cases returned for revision.
>
> We employed a hybrid human–LLM pipeline for privacy filtering. An LLM-based detector first automatically flagged potential personally identifiable information in screenshots and text fields. Flagged instances were then manually reviewed and redacted.
>
> The release contains screenshots, task instructions, action sequences, hierarchical UI documents, and box annotations, which are sufficient for fully reproducible offline training and evaluation without access to the original apps. GUIEvalKit further provides a self-contained evaluation pipeline operating entirely on this static data.
>
> We will add a dedicated appendix section describing annotation workflow, quality control, privacy filtering, and release constraints.
>
> ## For Concern 2
> We agree that the original scaling study, conducted on a single base model with one reward design, should be stated more carefully in scope. In the revision, we broaden the evidence along the two axes raised by the reviewer: model family and reward definition.
>
> We add a second scaling study on Qwen3-VL-8B-Thinking, a general-purpose VLM architecturally distinct from the GUI-specialized UI-TARS-1.5-7B. The results are consistent: RL outperforms SFT across data scales, and performance improves approximately log-linearly with training data size.
>
> We introduce a Gaussian spatial reward for click actions (which account for ~57% of test steps in HyperTrack), inspired by recent work on continuous reward modeling for GUI grounding (GUI-G²). Compared to the original binary hit/miss reward, the Gaussian formulation provides denser learning signals. Under this reward, RL exhibits a smoother learning curve while preserving the same scaling advantage over SFT.
>
> | Training Scheme \ Step | 16 | 32 | 128 | 256 | 512 | 1024 | 2048 | 4096 |
> | - | - | - | - | - | - | - | - | - |
> | Binary RL | 70.55 | 70.17 | 73.33 | 74.93 | 71.84 | 74.39 | 72.39 | 74.59 |
> | Gaussian RL | 68.62 | 68.18 | 71.68 | 71.60 | 71.65 | 72.59 | 74.51 | 70.66 |
> | SFT | 62.98  | 68.01 | 70.54 | 68.86 | 70.49 | 71.68  | 71.80  | 70.73 |
>
> These additions show that our core conclusions, log-linear scaling and RL's consistent advantage, are not artifacts of a single configuration, but hold across multiple base models and reward formulations. We note that our RL experiments use the DAPO recipe (a modern extension of GRPO/PPO with Clip-Higher, Dynamic Sampling, and token-level policy gradient). A systematic comparison across different RL algorithm families (e.g., PPO vs. GRPO vs. DAPO variants) is an important direction we leave for future work, and we will revise the paper to state the scope of our claims accordingly. Beyond our own experiments, we believe one key value of HyperTrack is enabling the community to conduct further scaling studies under diverse model families, RL algorithms, and reward designs. We will release the full training data and splits to facilitate such research.
>
> ## For Concern 3
> We apologize for the confusion. The screenshots in the appendix are drawn from the decision-level evaluation subset (Table 10), which spans all five benchmarks, including English-centric ones (AndroidControl, AiTZ, GUI Odyssey). This is why the displayed examples happen to be in English. The HyperTrack dataset itself is indeed collected from 674 Chinese Android apps with Chinese-language interfaces. In the revision, we will add representative Chinese-language screenshots from HyperTrack in the appendix.
>
> ## For Concern 4
> We agree that SOEval is not true online evaluation and does not fully capture trajectory-level error accumulation or long-horizon drift. We will state this limitation explicitly in the revision.
>
> Our design intent is narrower: SOEval is a step-level, context-aligned protocol that reduces the mismatch between offline evaluation (conditioned on reference history) and online inference (conditioned on model-generated history). It recovers on-policy context when the model stays on-track, and falls back to the reference trajectory otherwise. This is a deliberate trade-off — sacrificing full online fidelity for stable, fine-grained analysis of step-level decision behavior under a more realistic context distribution.
>
> For detailed quantitative evidence on its relation to online metrics, please see our **Response to Reviewer 1nSs**, _For Weakness 1, 3 and Question 1_.

---

> > ### Author Rebuttal · Reviewer_eVUL · 2026-04-01
> >
> > I have raised my score

---

> > > ### Author Response · Authors · 2026-04-03
> > >
> > > We are glad to see that we have addressed your concerns, and we sincerely appreciate your support in raising the score, as well as your constructive suggestions that helped improve our work.

---

### Decision · Program_Chairs · 2026-04-30

**Decision:**

Accept (regular)

**Comment:**

This paper provides a systematic study of VLM-based agents for mobile GUI navigation, focusing on data scaling, benchmarking, and reasoning behavior, introducing a large-scale Chinese dataset and a unified toolkit for both offline and semi-online evaluation. The initial ratings for this paper were split, with one weak reject, one strong reject, and two weak accepts. One of the reviewers who gave a weak reject raised concerns such as sensitivity to clustering and hyperparameters, episode-level success/progress metrics, and absent or limited evaluation. However, these concerns were well addressed in the rebuttal, leading this reviewer to raise the score to accept. The reviewer who initially gave a strong reject criticized many aspects of the paper—soundness, presentation, significance, and originality—as being poor; however, these criticisms lacked specificity and persuasiveness. This reviewer also noted that their understanding of the paper improved after reading the rebuttal and subsequently raised the score to weak reject. Respecting the majority opinion, the AC recommends acceptance of this paper.